# INTERACTIVE MULTI-EVENT VIDEO RETRIEVAL WITH CONTEXT INTEGRATION AND POSITION CONSTRAINT

## ABSTRACT

Interactive video retrieval aims to progressively refine queries through multi-round interactions between the user and the system. Existing methods focus on pre-trimmed videos that provide captions that well describe the gist of the video content. In real-world scenarios, however, videos typically contain a sequence of unrelated and discontinuous events, while a query usually refers to a single event. This mismatch introduces significant challenges, including sensitivity to irrelevant content, lack of context exploitation, and insufficient position exploration. Motivated by this, we propose **CIPC**, a tailored interactive video retrieval framework with Context Integration and Position Constraint for multi-event videos. CIPC adaptively segments videos into event-consistent units, supports progressive interactions that exploit contextual information, and incorporates a position constraint to re-weight candidate segments by temporal distance, promoting better temporal alignment with the query. Extensive experiments and a user simulation study demonstrate the effectiveness and robustness of our approach, yielding 4.1%–6.7% R@1 improvements on three widely used benchmarks.

## 1 INTRODUCTION

Text-video retrieval is a fundamental multi-modal task, aiming to retrieve the most semantically relevant video from database given a natural language description. Text-video retrieval (Wu et al., 2021; Luo et al., 2021; Liu et al., 2021) typically relies on the user's initial description, which is often a succinct query with semantic ambiguity (Zhang et al., 2025), thereby complicating retrieval. To this end, Interactive Video Retrieval (IVR) has attracted increasing research attention (Zhang et al., 2025; Han et al., 2024). As shown in Fig. 1 (a), an IVR system consists of two components: multi-round interactions between the user and the retrieval system to refine the semantics of original descriptions, and a retrieval system that utilizes the refined query to obtain more satisfactory results.

Interactive retrieval methods have been widely explored in various domains, including image retrieval, video retrieval, and person re-identification (Lu et al., 2025; Qin et al., 2025). For image retrieval, ChatIR (Levy et al., 2023) proposes the first chat-based image retrieval protocol, which progressively clarifies retrieval intent through dialogue with users and overcomes the limitations of traditional single-round text-to-image retrieval. In the video domain, MERLIN (Han et al., 2024) leverages cognitive feedback to simulate how humans iteratively refine queries through dialogue. Meanwhile, UMIVR (Zhang et al., 2025) approaches interaction optimization by quantifying three types of uncertainty, thereby enhancing the interaction process.

Current interactive video retrieval methods primarily focus on scenarios in which videos are pre-trimmed to ensure a strong semantic alignment with queries. However, as shown in Fig. 1 (b), real-world videos often encompass multiple events, whereas queries are typically specific and refer to a single event (Zhang et al., 2023). Directly applying existing interactive methods to multi-event videos faces the following challenges (Fig. 2): **(1) Sensitivity to irrelevant content.** Multi-event videos often consist of diverse segments, and treating the entire video as a whole not only reduces retrieval accuracy but also introduces irrelevant information and noise during the interaction stage. **(2) Lack of context exploitation.** Segments in multi-event videos typically exhibit temporal interrelations. Focusing solely on the target segment without leveraging temporal context results in an incomplete understanding of the query. For instance, given the segment *"He scrubs his hands with the soap"*, the subsequent segment is likely *"He turns on the faucet and rinses his hands and*

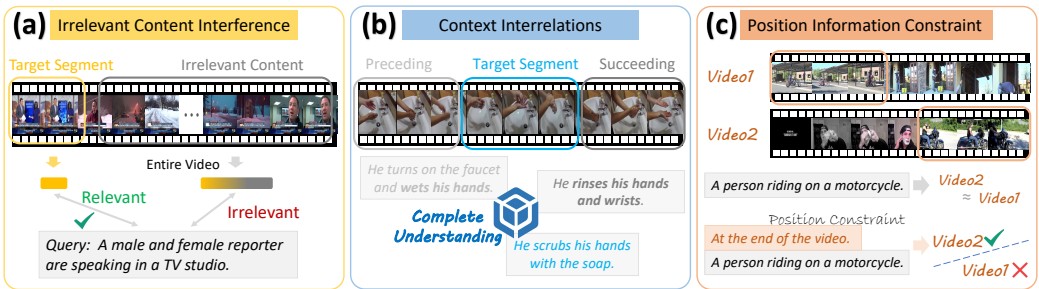

Figure 1: (a) Illustration of interactive video retrieval. Users often submit succinct queries that lack sufficient information to retrieve the desired videos. Interactive retrieval addresses this issue by encouraging users to provide more effective cues, enhancing the intelligence of the retrieval system, and improving the overall user experience. (b) The difference between pre-trimmed videos and multi-event videos. Pre-trimmed videos ensure high relevance between the query and entire videos, whereas multi-event videos contain multiple events, and queries typically refer to a single event.

Figure 2: Challenges in interactive multi-event video retrieval. (a) Treating the entire video as a whole makes the target segment susceptible to interference from irrelevant content. (b) The context understanding facilitates capturing more precise interrelations around the target segment. (c) Semantic retrieval may fail to differentiate videos with highly similar semantics but different temporal positions, while the incorporation of position constraints effectively compensates for this.

*wrists"*. **(3) Insufficient position exploration.** In multi-event videos, users are often concerned with the content at specific temporal positions. However, existing methods primarily focus on semantic information during interaction while neglecting temporal position cues. Intuitively, leveraging temporal position information can facilitate the discrimination of hard-negative candidates.

To address the aforementioned challenges, we leverage both contextual information and temporal positional cues to jointly guide the interactive retrieval process. We propose **CIPC**, a tailored interactive video retrieval framework with **C**ontext **I**ntegration and **P**osition **C**onstraint for multi-event videos, as shown in Fig. 3. Specifically, the proposed framework comprises three key modules: (1) Adaptive video segmentation. Videos are first adaptively segmented based on frame-wise differences. A coarse-to-fine strategy is then employed: coarse-grained segmentation captures prominent event boundaries or scene changes, followed by fine-grained segmentation to distinguish detailed changes within each segment. (2) Progressive interaction with context integration. During interaction, users provide both broad descriptions for coarse segments and detailed descriptions for fine segments. Furthermore, a context-aware mechanism enables both segment-specific refinement and context-integrated refinement. (3) Position constraint. Temporal positions of queries are leveraged to re-weight candidate segments according to their temporal proximity to the query, encouraging better alignment and improving retrieval accuracy in multi-event videos.

Our main contributions are summarized as follows: (1) We analyze the limitations of existing interactive video retrieval methods on real-world untrimmed videos. To the best of our knowledge, this work is the first systematic study to consider the unique characteristics and challenges of interactive untrimmed video retrieval. (2) We propose a tailored interactive retrieval framework with adaptive video segmentation, progressive interaction with context integration, and position constraint, effectively addressing the challenges posed by multi-event videos. (3) Extensive experiments on three challenging benchmarks demonstrate that our method outperforms both interactive systems and non-interactive baselines, and ablation studies validate the effectiveness of each proposed component.

## 2 Related Work

### 2.1 Interactive Retrieval

Unlike traditional retrieval approaches, interactive retrieval emphasizes the importance of user participation to improve retrieval results, which allows users to progressively refine search results through continuous interaction with the retrieval system. Early studies propose a paradigm of iteratively updating query representations based on user feedback through relevance feedback (Rocchio Jr, 1971; Salton & Buckley, 1990; Rui et al., 1998). In recent years, interactive retrieval has been increasingly applied to images, videos, and multimodal scenarios driven by the progress of Large Language Models (LLMs). ChatIR (Levy et al., 2023) proposes a new Chat-based Image Retrieval protocol. They propose a dialogue-based approach that uses large language models to generate follow-up questions, enabling to processing of the emerged dialogue into improved retrieval results. PlugIR (Lee et al., 2024) identifies the limitation of zero-shot models to understand dialogues and introduces an LLM questioner to address the searching bottleneck. They also propose a new evaluation metric tailored for interactive retrieval. In the Person Re-Identification (ReID) community, interactive dialogue has also gained attention. LLaVA-ReID (Lu et al., 2025) introduces a selective multi-image questioner that generates targeted clarification questions to refine witness descriptions. Similarly, Qin et al. (2025) leverages multi-round multimodal large language models interactions to enrich supervision and improve cross-modal matching. These works demonstrate the effectiveness of dialogue-driven refinement in ReID settings. For video retrieval, Han et al. (2025) addresses the limitation of one-way interaction in existing video retrieval systems and proposes a new task called Interactive Video Corpus Retrieval. They also design the InterLLaVA framework, which balances efficiency and accuracy via a two-stage retrieval process and produces interpretable results. MERLIN (Han et al., 2024) addresses a critical gap in the field of text-video retrieval and utilizes LLMs for iterative feedback learning. UMIVR (Zhang et al., 2025) proposes an uncertainty-aware interactive framework that systematically quantifies and minimizes three fundamental uncertainties–text ambiguity, mapping uncertainty, and frame uncertainty. These works encounter several challenges in a multi-event video scenario (Zhang et al., 2023).

In this paper, we focus on this and introduce a dedicated context integration and position constraint method for interactive multi-event video retrieval.

### 2.2 Multi-event Video-Text Retrieval

Multi-event Video-Text Retrieval (MeVTR) (Zhang et al., 2023) refers to retrieving untrimmed videos that usually involve multiple events. Unlike traditional video retrieval tasks that assume full relevance between videos and queries, MeVTR emphasizes that videos are generally untrimmed, containing a sequence of unrelated and discontinuous events, and a query typically refers to a single event. Zhang et al. (2023) introduces a simple model incorporating key event video representation and proposes a new multi-event loss for the task. Some other works are also dedicated to improving this task, typically named partially relevant video retrieval. MS-SL (Dong et al., 2022) treats the video as a bag of clips and frames, and jointly learns clip-level and frame-level similarities. GMMFormer (Wang et al., 2024b) uses a Gaussian Mixture Model to cluster video frames and efficiently model partially relevant segments, while GMMFormer v2 (Wang et al., 2024a) adds uncertainty-aware mechanisms to further improve retrieval robustness and efficiency. Dong et al. (2023) proposes the DL-DKD method, which transfers knowledge from the CLIP model through dynamic knowledge distillation. Additional studies focus on improving the alignment between textual queries and video content (Jiang et al., 2023; Chen et al., 2023). In contrast to these non-interactive methods, we investigate improving multi-event video retrieval by a practical interaction approach.

## 3 Method

The illustration of CIPC is shown in Fig. 3. We begin with introducing the overview of CIPC in Sec. 3.1, then increasingly present our modules of adaptive video segmentation in Sec. 3.2, progressive interaction with context integration in Sec. 3.3, and semantics-position co-retrieval in Sec. 3.4.

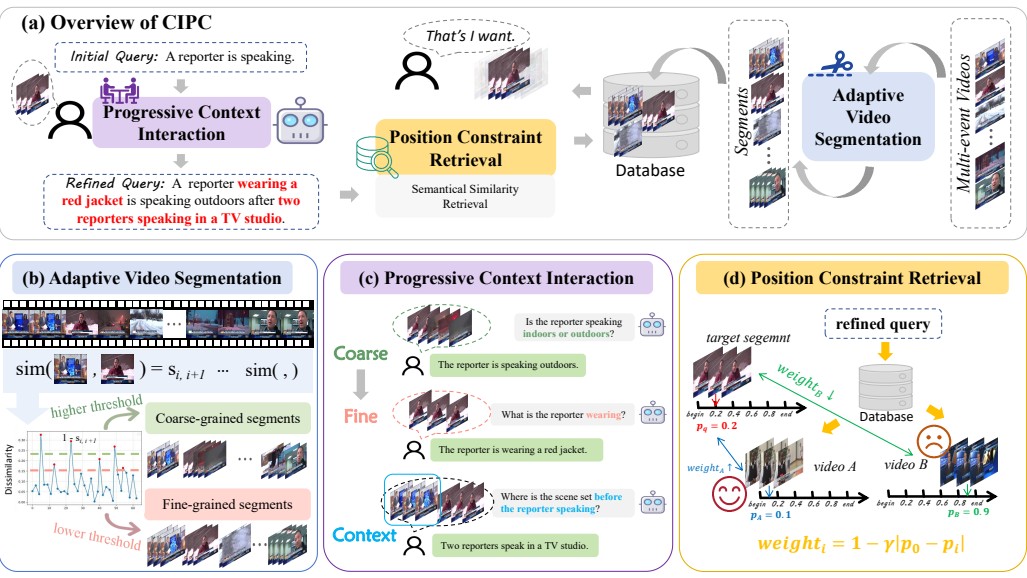

Figure 3: (a) Overview of our interactive retrieval framework with context integration and position constraint (CIPC) for multi-event videos. Multi-event videos are adaptively segmented and stored in the database. During retrieval, users' initial queries are progressively refined using our proposed coarse-to-fine interaction. The refined query is employed to retrieve the desired video from the database using our semantics-position co-retrieval strategy. (b), (c) and (d) are detailed illustrations of adaptive video segmentation, progressive context interaction, and position constraint retrieval.

## 3.1 OVERVIEW

As illustrated in Fig. 3 (a), we target the realistic multi-event video scenario and a tailored interactive video retrieval framework (CIPC) with Context Integration and Position Constraint. CIPC first adopts the proposed adaptive video segmentation to split multi-event videos into several segments and stores them in a database for subsequent interaction and retrieval. Typically, the initial query is usually vague and lacks sufficient details, such as *"A reporter is speaking"*. CIPC refines the initial query by supplementing more details through progressive context interaction, such as the reporter's gender and appearance. After the interaction, the query is refined to *"A reporter **wearing a red jacket** is speaking outdoors after **two reporters speaking in a TV studio**"*. The refined query is then used to perform both position constraint retrieval and semantic retrieval, retrieving the video with the desired segment from the database. The main modules of adaptive video segmentation, progressive context interaction, and position constraint retrieval are elaborated in the following sections.

## 3.2 ADAPTIVE VIDEO SEGMENTATION

Given a set of videos $\mathcal{V}$ and a set of queries $\mathcal{Q}$, we first uniformly sample $F$ frames from each video $v \in \mathcal{V}$ and extract frame-level features using a vision encoder. Consequently, a video $v$ is represented as $\mathbf{e}_v = \{\mathbf{e}_v^1, \mathbf{e}_v^2, \ldots, \mathbf{e}_v^F\} \in \mathbb{R}^{F \times d}$, where $d$ denotes the feature dimension. Since untrimmed videos often consist of diverse segments, we propose an adaptive video segmentation to split each video into semantically coherent segments. Specifically, as shown in Fig. 3 (b), we first compute the cosine similarity between consecutive frames:

$$s_v^j = \text{sim}(\mathbf{e}_v^j, \mathbf{e}_v^{j+1}), \quad j = 1, 2 \ldots, F-1, \tag{1}$$

and derive dissimilarity scores $\delta_j = 1 - s_v^j$. An adaptive threshold $\tau$ is then calculated based on the mean $\mu_\delta$ and standard deviation $\sigma_\delta$ of the dissimilarities:

$$\tau = \mu_\delta + k \cdot \sigma_\delta, \tag{2}$$

where $k$ controls the sensitivity of the segmentation. Frame $j$ is recognized as a potential cut point whenever $\delta_j > \tau$. The final segmentation is denoted by

$$\mathcal{S} = \{(b_1, e_1), (b_2, e_2), \ldots, (b_M, e_M)\}, \tag{3}$$

where $(b_m, e_m)$ denote the start and end frames of the $m$-th segment, and $M$ is the number of segments. Finally, we compute segment-level features $\mathbf{e}_s^i$ by averaging the frame features within the $i$-th segment, yielding the set of segment representations $\mathbf{e}_s = \{\mathbf{e}_s^1, \mathbf{e}_s^2, \ldots, \mathbf{e}_s^M\} \in \mathbb{R}^{M \times d}$.

## 3.3 PROGRESSIVE INTERACTION WITH CONTEXT INTEGRATION

**Coarse-to-Fine Interaction.** To facilitate more effective interaction, we adopt a coarse-to-fine segmentation strategy. Specifically, we first set a relatively large threshold (higher hyperparameter $k$), leading to fewer cut points and longer segments. This coarse-grained segmentation captures prominent event boundaries or scene changes. Subsequently, each coarse-grained segment can be further divided using a smaller threshold by a lower $k$ to achieve fine-grained segmentation, helping to distinguish detailed changes within the segment.

As shown in Fig. 3 (c), at the coarse-grained stage, the system VLM intends to generate broad clarifying questions, such as *"Is the scene indoors or outdoors?"*, which helps identify prominent event boundaries and progressively narrows down the scope to the target segment. At the fine-grained stage, the system VLM shifts to more detailed and localized clarifying questions, such as *"What is the man holding?"* or *"What color is her shirt?"*, which refine subtle semantic cues. In this way, the search space is narrowed, guiding the interaction from coarse to fine granularity.

To further improve the efficiency during multi-round interactions, we introduce an early stop strategy. Since query refinement converges once no new details can be provided, continuing the dialogue becomes redundant. Concretely, at interaction round $r$, we encode the current refined query $q_r$ and the query from the previous round $q_{r-1}$ into vectors $\mathbf{e}_t^r$ and $\mathbf{e}_t^{r-1}$ using a text encoder, and compute their cosine similarity $s_t^r = \text{sim}(\mathbf{e}_t^r, \mathbf{e}_t^{r-1})$. If consecutive refined queries become highly similar, the interaction is terminated, indicating that the query has converged.

**Interaction with Context Integration.** In multi-event videos, segments typically exhibit temporal continuity and strong interrelations. Considering only the target segment is insufficient, since context segments provide valuable supplementary information. To address this, our framework explicitly integrates context information into the interaction process.

In the early rounds, clarifying questions are generated solely based on the target segment to ensure precise refinement. In later rounds, the process bifurcates into two branches: one continues segment-specific refinement, while the other introduces context-integrated refinement by generating and answering clarifying questions regarding preceding or succeeding segments. This design enables the refined query to capture both localized details and broader contextual cues.

## 3.4 SEMANTICS & POSITION CO-RETRIEVAL

In addition to the widely studied semantic retrieval, we introduce position constraint retrieval, which operates collaboratively with semantic alignment to accurately retrieve the desired video.

### 3.4.1 SEMANTICS RETRIEVAL

In multi-event videos, segments often exhibit diverse semantics and strong interrelations. To fully exploit this information, we leverage both segment-specific refinement and context-integrated refinement generated during the interaction process. Formally, given the segment features $\mathbf{e}_s = \{\mathbf{e}_s^1, \mathbf{e}_s^2, \ldots, \mathbf{e}_s^M\}$, we denote the segment-specific and context-integrated query refinements as $\mathbf{e}_{t,s}$ and $\mathbf{e}_{t,c}$, respectively. The segment-specific similarity is computed directly using cosine similarity:

$$s_s^i = \text{sim}(\mathbf{e}_{t,s}, \mathbf{e}_s^i), \quad i = 1, \ldots, M. \tag{4}$$

To capture contextual information, we first apply a local smoothing operation on segment features, equivalent to a 1D convolution with a fixed kernel $\left[\frac{1-\beta}{2}, \beta, \frac{1-\beta}{2}\right]$, which integrates information from contextual segments:

$$\hat{\mathbf{e}}_s^i = \tfrac{1-\beta}{2} \mathbf{e}_s^{i-1} + \beta \mathbf{e}_s^i + \tfrac{1-\beta}{2} \mathbf{e}_s^{i+1}, \quad i = 1, \ldots, M, \tag{5}$$

where $\beta \in [0, 1]$ balances the contribution of the target segment and its context. The context-integrated similarity is then computed as:

$$s_c^i = \text{sim}(\mathbf{e}_{t,c}, \hat{\mathbf{e}}_s^i). \quad i = 1, \ldots, M. \tag{6}$$

### 3.4.2 POSITION CONSTRAINT

We emphasize that segments not only contain rich semantic content but also occupy explicit temporal positions. During interaction, users may provide coarse information about the approximate location of the target segment within the video, which can serve as a useful cue for retrieval. To explore the effectiveness of positional guidance, we introduce a position constraint to leverage such information. Specifically, as shown in Fig. 3 (d), in the database, each candidate segment $(b_i, e_i)$ is assigned a normalized center position:

$$p_i = \frac{b_i + e_i}{2F}. \tag{7}$$

The temporal alignment between the query and each segment is encoded as a weighting factor:

$$w_i = 1 - \gamma |p_i - p_q|, \tag{8}$$

where $p_q$ denotes the query's normalized temporal position, and $\gamma \in [0, 1]$ controls the decay rate. $w_i$ modulates the semantic similarity scores obtained from segment-specific and context-integrated refinements, resulting in a weighted similarity for each segment. The overall video-level similarity is then computed by taking the maximum weighted similarity across all segments:

$$s_{\text{video}} = \max_{i=1}^{M} w_i \left( \alpha \, s_s^i + (1 - \alpha) \, s_c^i \right), \tag{9}$$

where $\alpha \in [0, 1]$ controls the balance between segment-specific and context-integrated refinement. The $s_{\text{video}}$ is finally employed to rank the candidate videos. By incorporating this position constraint, the retrieval process favors segments that are temporally aligned with the query, effectively exploiting the temporal position information that users may provide during the interaction.

There are multiple ways to simulate the query's normalized temporal position. For simplicity, we simulate it by the similarity computation. Let $\mathbf{e}_t$ denote the initial query feature and $\tilde{\mathbf{e}}_v^i$ denote the feature of the $i$-th frame in the target video with $F$ frames, we define the query's normalized temporal position using cosine similarity:

$$p_q = \frac{\arg\max_{i=1}^{F} \text{sim}(\mathbf{e}_t, \tilde{\mathbf{e}}_v^i)}{F}. \tag{10}$$

## 4 EXPERIMENTS

### 4.1 EXPERIMENTAL SETTINGS

**Dataset.** We verify our approach on three commonly used multi-event datasets of ActivityNet Captions (Krishna et al., 2017), Charades-STA (Gao et al., 2017), and TV show Retrieval (TVR) (Lei et al., 2020). **ActivityNet Captions** is a large-scale dataset consisting of 20K videos amounting to 849 video hours with 100K total descriptions taken from ActivityNet (Heilbron et al., 2015). On average, each video has around 3.7 moments with sentence descriptions. We follow the test setup in Escorcia et al. (2019) with 4,917 videos and 17,505 texts. **Charades-STA** contains 6,670 videos with 16,128 descriptions. Each video has roughly 3.7 moments on average. We follow the test set in Zhang et al. (2023) comprising 1,334 videos with 3,720 texts. **TVR** contains 109K queries collected from 21.8K videos across 6 TV shows of diverse genres, and each video has 5 moments on average. There are 2,179 videos with 10,895 texts in the test set following Zhang et al. (2020).

**Evaluation Metric.** In addition to the commonly used metric Recall at rank K (R@K) in retrieval evaluation, we adopt two metrics widely used in interactive retrieval: Hit at rank K (Hit@K) and Best log Rank Integral (BRI) (Lee et al., 2024). **R@K** measures the percentage of test samples where the target appears within the top-K retrieved results. **Hit@K** indicates whether the target appears within the top-K retrieved results during any interaction step. **BRI** is proposed by Lee et al. (2024) to evaluate an interactive retrieval system in three aspects: user satisfaction, efficiency, and ranking improvement significance. Higher R@K or Hit@K and lower BRI indicate better performance.

**Implementation Details.** For all datasets, we uniformly sample $F = 64$ frames from each video and extract frame-level features as well as encode textual queries using the vision and text encoders of CLIP Vit-B/32 (Radford et al., 2021). In all experiments (unless otherwise specified), we adopt

Table 1: Comparison with interactive video retrieval methods on three typical datasets. Retrieval performances of multiple rounds are reported for in-depth comparison. Our CIPC consistently surpasses the vanilla method and previous interaction methods across different interaction rounds.

| Method | Rounds | ActivityNet Captions | | | | Charades-STA | | | | TVR | | | |
|---|---|---|---|---|---|---|---|---|---|---|---|---|---|
| | | R@1↑ | R@10↑ | Hit@1↑ | BRI↓ | R@1↑ | R@10↑ | Hit@1↑ | BRI↓ | R@1↑ | R@10↑ | Hit@1↑ | BRI↓ |
| UMIVR (Zhang et al., 2025) | 0 | 9.9 | 34.6 | 9.9 | - | 1.1 | 6.6 | 1.1 | - | 5.6 | 19.9 | 5.6 | - |
| | 3 | 18.9 | 54.9 | 27.3 | 2.51 | 3.7 | 18.0 | 6.7 | 4.24 | 9.4 | 30.2 | 16.0 | 3.44 |
| | 7 | 22.1 | 58.2 | 32.1 | 2.13 | 8.6 | 26.4 | 15.5 | 3.61 | 11.0 | 35.6 | 21.5 | 3.02 |
| | 10 | 23.6 | 59.1 | 33.1 | 2.01 | 9.6 | 30.2 | 16.8 | 3.52 | 14.2 | 41.7 | 25.8 | 2.77 |
| MERLIN (Han et al., 2024) | 0 | 9.9 | 34.6 | 9.9 | - | 1.1 | 6.6 | 1.1 | - | 5.6 | 19.9 | 5.6 | - |
| | 3 | 19.3 | 55.9 | 29.1 | 2.34 | 4.6 | 18.7 | 6.4 | 4.16 | 12.5 | 34.1 | 19.3 | 3.26 |
| | 7 | 23.2 | 63.0 | 36.1 | 2.08 | 9.7 | 27.8 | 16.6 | 3.67 | 14.7 | 40.2 | 25.0 | 2.84 |
| | 10 | 25.8 | 64.9 | 38.2 | 1.95 | 10.5 | 34.7 | 17.1 | 3.49 | 16.1 | 43.2 | 27.3 | 2.61 |
| Vanilla Method | 0 | 9.9 | 34.6 | 9.9 | - | 1.1 | 6.6 | 1.1 | - | 5.6 | 19.9 | 5.6 | - |
| | 3 | 17.6 | 52.5 | 23.4 | 2.49 | 3.6 | 19.6 | 4.8 | 4.32 | 7.5 | 25.7 | 11.5 | 3.67 |
| | 7 | 20.1 | 56.2 | 28.4 | 2.10 | 6.5 | 21.8 | 11.2 | 3.81 | 8.4 | 27.5 | 15.3 | 3.31 |
| | 10 | 22.1 | 57.9 | 32.0 | 1.98 | 9.2 | 26.3 | 14.7 | 3.60 | 10.1 | 29.4 | 18.0 | 3.16 |
| CIPC (ours) | 0 | 11.8 (+1.9) | 36.1 (+1.5) | 11.8 (+1.9) | - | 1.7 (+0.6) | 7.6 (+1.0) | 1.7 (+0.6) | - | 9.0 (+3.4) | 30.4 (+10.5) | 9.0 (+3.4) | - |
| | 3 | 20.3 (+1.0) | 56.8 (+0.9) | 28.1 (-1.0) | 2.23 (-0.11) | 4.3 (-0.3) | 21.2 (+1.6) | 6.9 (+0.2) | 4.05 (-0.11) | 17.0 (+4.5) | 49.5 (+15.4) | 25.1 (+5.8) | 2.70 (-0.56) |
| | 7 | 27.9 (+4.7) | 66.5 (+3.5) | 39.9 (+3.8) | 1.80 (-0.28) | 10.3 (+0.6) | 32.4 (+4.6) | 16.2 (-0.4) | 3.52 (-0.15) | 20.8 (+6.1) | 58.1 (+18.1) | 33.4 (+8.4) | 2.38 (-0.46) |
| | 10 | **31.3** (+5.5) | **69.3** (+4.4) | **45.1** (+6.9) | **1.62** (-0.33) | **14.6** (+4.1) | **43.3** (+8.3) | **23.5** (+6.4) | **3.21** (-0.28) | **22.8** (+6.7) | **60.9** (+17.7) | **36.4** (+9.1) | **2.24** (-0.37) |

*Doubao-Seed-1.6* (Guo et al., 2025) as the system model for generating questions and refining queries with a temperature of 0.1, and use Qwen2.5-VL-7B (Bai et al., 2025) as the user model for answering with a temperature of 0.7. During the interaction process, we set $k = 2$ for coarse-grained segmentation and $k = 1$ for fine-grained segmentation, with a maximum of 10 rounds. We use a fixed schedule: 4 coarse-grained rounds followed by 6 fine-grained rounds, balancing efficiency and accuracy. And during the 10-round interaction, one branch always refines the query on the target segment, while the other switches after round 8 to include preceding and succeeding segments for context. At the retrieval stage, we use default hyperparameters of $\alpha = 0.3$, $\beta = 0.9$, and $\gamma = 0.3$.

**Vanilla Method.** For the vanilla method, we do not perform any segmentation and instead treat each video as a whole. During interaction, no specialized interaction strategy is adopted. At each round, the system generates clarification questions and provides answers solely based on the query, while keeping the procedure identical across all rounds. Other settings remain consistent with CIPC.

## 4.2 COMPARISON RESULTS

**Comparison with Interactive Video Retrieval.** To compare with existing interactive video retrieval approaches in more realistic scenarios, we adapt UMIVR (Zhang et al., 2025) and MERLIN (Han et al., 2024) to multi-event video datasets. Since the official implementation of UMIVR is not publicly available, we report the results reproduced based on the descriptions in their paper. As shown in Tab. 1, these methods are originally designed for pre-trimmed videos fully relevant to the query, limiting their performance on multi-event datasets. For instance, on ActivityNet Captions, UMIVR achieves an R@1 of 23.6 after 10 rounds, slightly above the vanilla method. Across all methods, increasing interaction rounds consistently improves retrieval performance. For example, the R@1 of MERLIN on Charades-STA improves from 1.1 to 10.5 after 10 rounds, showing that iterative feedback helps the model refine its understanding of the user query and video content. Our CIPC consistently outperforms all methods across all datasets after 10 rounds. Notably, it achieves R@1 of 31.3, 14.6, and 22.8 on three datasets, respectively, showing that CIPC is more effective in multi-event scenarios. Moreover, CIPC achieves lower BRI values, indicating higher efficiency.

**User Simulation Study.** In the paradigm of interactive retrieval (Lee et al., 2024; Han et al., 2024; Zhang et al., 2025), a VLM is typically used to simulate a real-world user to answer clarifying questions proposed by the retrieval system. Generally, user VLMs leverage the ground truth video as input, which creates an idealized experimental setting that rarely holds in real-world scenarios. Users may forget specific details, overlook relevant context, or provide inaccurate descriptions due to subjective impressions. Considering this, we conduct comprehensive experiments to simulate the real-world scenarios of imperfect user feedback, verifying the robustness of CIPC. Specifically, we inject Gaussian noise into video frames, controlling the standard deviation $\sigma$ of the Gaussian noise to adjust the noise level and thereby reduce the amount of information directly accessible to the user VLM. In our experiments, we consider two noise levels, setting $\sigma = 20$ and $\sigma = 50$, which

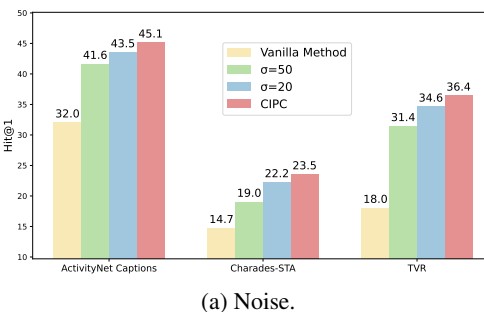 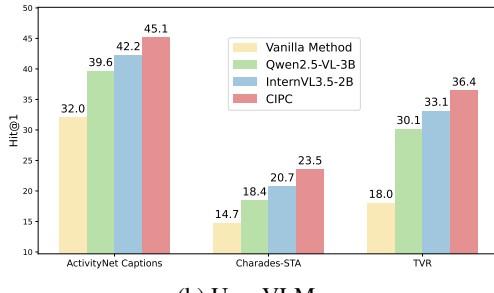

|            (a) Noise.            |            (b) User VLMs.            |

Figure 4: User simulation study. We consider two situations to simulate different response accuracy provided by the user. (a) Noise: Gaussian noise is injected into the video, where a larger $\sigma$ represents more forgetting and uncertainty of the user response. (b) User VLMs: VLMs of different capabilities are employed to simulate diverse user responses, where the smaller models are used to simulate low-quality response scenarios. CIPC exhibits effectiveness even if the user's answer is not accurate.

Table 2: Evaluation of noisy or misattributed temporal hints on ActivityNet Captions dataset.

| Noise | R@1↑ | R@10↑ | Hit@1↑ | BRI↓ |
|---|---|---|---|---|
| 10% | 31.1 | **69.3** | 44.7 | 1.64 |
| 20% | 30.6 | 68.6 | 44.3 | 1.64 |
| 50% | 29.3 | 67.6 | 42.2 | 1.70 |
| CIPC (ours) | **31.3** | **69.3** | **45.1** | **1.62** |

correspond to moderate and severe information degradation, respectively. What's more, we also employ smaller-scale user VLMs, such as InternVL3.5-2B (Wang et al., 2025) and Qwen2.5-VL-3B (Bai et al., 2025), to simulate low-quality response scenarios. The results are presented in Fig. 4. Under both noise settings ($\sigma = 20$ and $\sigma = 50$), our method exhibits a performance drop across three datasets, but still significantly outperforms the vanilla method. Moreover, when employing smaller-scale VLMs to provide low-quality feedback, our approach remains effective.

In position constraint, users may provide only coarse or noisy temporal hints. To study the robustness of our method under such conditions, we conduct additional experiments by adding noise to simulate imperfect user guidance. To be specific, we add Gaussian noise to video frame features before computing query–frame similarity. The noise levels (10%, 20%, 50%) indicate increasing noise strength, simulating noisier or misattributed user hints. Results are shown in Tab. 2. Performance decreases gradually as noise increases. However, the degradation is relatively small, demonstrating strong robustness to noisy temporal hints.

**Comparison with Non-interactive Video Retrieval.** We compare our method with existing non-interactive approaches, which focus on multi-event video retrieval. Unlike interactive retrieval, these methods directly retrieve target videos based on a single query without iterative user feedback. The results are reported in Tab. 3. These methods achieve limited performance, indicating that without iterative feedback, non-interactive approaches struggle to retrieve relevant events in untrimmed videos accurately. CIPC quickly surpasses these methods after only two rounds, reaching an R@1 of 19.1 on ActivityNet Captions. With more interaction rounds, the performance gap further enlarges.

### 4.3 ABLATION STUDY

**Key Components.** In this section, we conduct ablation experiments to verify the contribution of each component in our framework, as shown in Tab. 4. Adaptive segmentation significantly improves retrieval performance by selecting informative clips and reducing irrelevant content interference, as seen by the increase from 22.1 to 26.0 in R@1. Segment-specific refinement helps reduce redundancy, and context-integrated refinement enhances contextual understanding. Combining both achieves the best results. Position constraint further boosts R@1 and Hit@1 while lowering BRI,

Table 3: Comparison with non-interactive multi-event video retrieval methods on three datasets. Methods marked with † utilize CLIP ViT-B/32 as a backbone, while others use ResNet. Our CIPC quickly surpasses advanced non-interactive methods, highlighting the effectiveness of interaction.

| Method | ActivityNet Captions | | | Charades-STA | | | TVR | | |
|---|---|---|---|---|---|---|---|---|---|
| | R@1↑ | R@10↑ | Hit@1↑ | R@1↑ | R@10↑ | Hit@1↑ | R@1↑ | R@10↑ | Hit@1↑ |
| *Non-interactive methods* | | | | | | | | | |
| MS-SL (Dong et al., 2022) | 7.1 | 34.7 | 7.1 | 1.8 | 11.8 | 1.8 | 13.5 | 43.4 | 13.5 |
| JSG (Chen et al., 2023) | 6.8 | 34.8 | 6.8 | 2.4 | 12.8 | 2.4 | - | - | - |
| PEAN (Jiang et al., 2023) | 7.4 | 35.5 | 7.4 | 2.7 | 13.5 | 2.7 | 13.5 | 44.1 | 13.5 |
| DL-DKD (Dong et al., 2023) | 8.0 | 37.5 | 8.0 | - | - | - | 14.4 | 45.8 | 14.4 |
| GMMFormer (Wang et al., 2024b) | 8.3 | 36.7 | 8.3 | 2.1 | 12.5 | 2.1 | 13.9 | 44.5 | 13.9 |
| MeVTR† (Zhang et al., 2023) | 16.0 | 49.6 | 16.0 | 2.0 | 12.7 | 2.0 | - | - | - |
| *Interactive methods* | | | | | | | | | |
| CIPC† (round 0) | 11.8 | 36.1 | 11.8 | 1.7 | 7.6 | 1.7 | 9.0 | 30.4 | 9.0 |
| CIPC† (round 2) | 19.1 | 54.7 | 24.5 | 3.2 | 16.6 | 5.1 | 16.0 | 47.4 | 22.6 |
| CIPC† (round 10) | **31.3** | **69.3** | **45.1** | **14.6** | **43.3** | **23.5** | **22.8** | **60.9** | **36.4** |

Table 4: Ablations of key components of CIPC on the ActivityNet Captions dataset. "Segment" and "Context" represent segment-specific and context-integrated refinement in interaction, respectively.

| Adaptive Segmentation | Interaction | | Position Constraint | R@1↑ | Hit@1↑ | BRI↓ |
|---|---|---|---|---|---|---|
| | Segment | Context | | | | |
| ✗ | ✗ | ✗ | ✗ | 22.1 | 32.0 | 1.98 |
| ✓ | ✓ | ✓ | ✗ | 26.0 | 41.6 | 1.70 |
| ✓ | ✓ | ✗ | ✓ | 28.0 | 43.4 | 1.64 |
| ✓ | ✗ | ✓ | ✓ | 30.7 | 44.3 | 1.63 |
| ✓ | ✓ | ✓ | ✓ | **31.3** | **45.1** | **1.62** |

indicating its effectiveness in reducing irrelevant matches. Overall, each component contributes positively, and the full CIPC framework achieves the highest performance.

**Video Segmentation Methods.** We evaluate different segmentation strategies, e.g., uniform segmentation, K-means clustering, and sliding windows on ActivityNet Captions, to compare with our adaptive video segmentation. From Tab. 5, we observe that our adaptive segmentation consistently outperforms other methods, achieving higher R@1 and Hit@1 while maintaining the lowest BRI, demonstrating its effectiveness in segmenting event-consistent units and selecting informative video segments.

Table 5: Ablations on segmentation methods.

| Method | Rounds | R@1↑ | Hit@1↑ | BRI↓ |
|---|---|---|---|---|
| Uniform | 10 | 29.6 | 43.4 | 1.68 |
| K-means | 10 | 28.9 | 42.9 | 1.75 |
| Sliding windows | 10 | 30.1 | 44.2 | 1.70 |
| Adaptive (ours) | 10 | **31.3** | **45.1** | **1.62** |

**Different Questioners.** We study the effect of different questioners, including different question generation models and strategies on Charades-STA. For different models, we adopt commercial models like GPT-4o-latest and QVQ-Max, and the open-source model Qwen2.5-VL-7B and Qwen3-8B (Yang et al., 2025). As shown in Fig. 5a, more advanced commercial models generate more informative questions and achieve better results, while Qwen2.5-VL-7B, with weaker language ability, is less effective. However, when using the newer open-source Qwen3-8B as the system model, performance is quite strong, with only a small gap compared to the commercial models. We also investigate three strategies for generating questions: (i) based solely on the query, (ii) with respect to the top-5 retrieved videos each round, and (iii) generating all 10 questions at once. Fig. 5b shows that strategy (ii) slightly improves performance

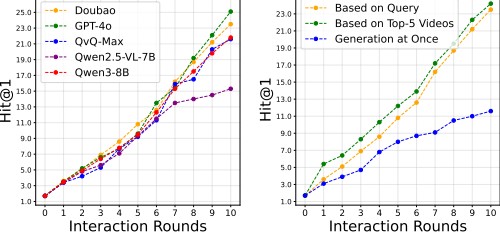

(a) Different models.    (b) Different strategies.

Figure 5: Ablations on different questioners.

but is computationally inefficient, while strategy (iii) prevents the model from adapting questions based on previous answers, limiting effective refinement in multi-round interactions. On ActivityNet Captions, generating questions per round with the top-5 retrieval strategy takes 8.14s and 30.7GB of GPU memory, compared to 0.55s and 15.7GB for our CIPC method.

**Hyper-parameters.** We analyze the sensitivity of three key hyperparameters in the retrieval stage (Sec. 3.4): $\alpha$, governing the fusion ratio between segment-specific and context-integrated similarities; $\beta$, balancing the target segment and its context during smoothing; and $\gamma$, controlling the temporal decay rate for positional weighting. Results are presented in Fig. 6. Ablation on $\alpha$ shows the best performance at $\alpha = 0.3$. The fact that $\alpha = 1$ performs worse than $\alpha = 0$ highlights the importance of context-integrated refinement. Performance with $\beta$ peaks at 0.9, indicating that lower values overemphasize context-integrated refinement while higher values give excessive weight to segment-specific refinement, highlighting the need for a balanced weighting. For $\gamma$, performance peaks at 0.3. Lower values reduce the influence of temporal position, weakening alignment with the query, while higher values overweight temporal distance, potentially ignoring semantic similarity.

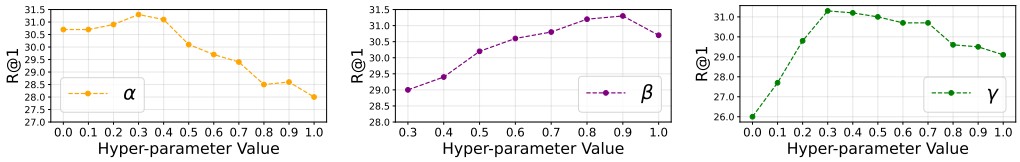

Figure 6: Influence of the three hyper-parameters.

## 5 CONCLUSION

In this paper, we focus on the interactive retrieval for real-world scenarios where video content usually encompasses multiple events, which brings about critical challenges: sensitivity to irrelevant content, lack of context exploitation, and insufficient position exploration. To this end, we propose an interactive retrieval framework for multi-event videos that integrates adaptive video segmentation, progressive interaction with context integration, and position constraint. Extensive experiments on benchmarks demonstrate that our method outperforms both interactive video retrieval approaches and non-interactive baselines, validating its effectiveness in handling complex, multi-event video content. To simulate real-world scenarios where users provide incomplete or ambiguous descriptions during interaction, we further conduct experiments by adding Gaussian noise to video frames and choosing smaller-scale models, demonstrating our method's robustness.

## 6 ETHICS STATEMENT

This work adheres to the ICLR Code of Ethics. In this study, no human subjects or animal experimentation were involved. All datasets used were sourced in compliance with relevant usage guidelines, ensuring no violation of privacy. We have taken care to avoid any biases or discriminatory outcomes in our research process. No personally identifiable information was used, and no experiments were conducted that could raise privacy or security concerns. We are committed to maintaining transparency and integrity throughout the research process.

## 7 REPRODUCIBILITY STATEMENT

We have made every effort to ensure that the results presented in this paper are reproducible. All code and datasets have been made publicly available if the paper is accepted to facilitate replication and verification. The experimental setup, such as model configurations, is described in detail in the paper to assist others in reproducing our experiments. Besides, the datasets used in the paper are publicly available, ensuring reproducible evaluation results.

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

# A APPENDIX

We present supplementary content in this appendix, organized as follows: Section A.1 describes detailed results of the user simulation study; Section A.2 illustrates the concrete interactive retrieval process; Section A.3 shows an example illustrating how noise can lead to interaction failures; Section A.4 supplements ablation studies on different design choices and experimental settings. Section A.5 presents the prompts used for the system VLM and the user VLM.

## A.1 USER SIMULATION STUDY

For intuitive presentation of the results, the main text employs visualized bar charts. Here, we provide more detailed quantitative results of this experiment as a supplement as shown in Tab. 6.

Table 6: User simulation study. We experiment with two approaches to simulate different response accuracy provided by the user. Noise: Gaussian noise is injected into the video, where larger $\sigma$ represents more forget and uncertainty of user response. User VLMs: VLMs of different capabilities are employed to simulate diverse user response, where smaller 2B model is used to simulate low-quality response scenarios. Our method exhibits effective even if the user's answer is not accurate.

| Type | Setting | Rounds | ActivityNet Captions | | | Charades-STA | | | TVR | | |
|------|---------|--------|-------|--------|------|-------|--------|------|-------|--------|------|
| | | | R@1↑ | Hit@1↑ | BRI↓ | R@1↑ | Hit@1↑ | BRI↓ | R@1↑ | Hit@1↑ | BRI↓ |
| Noise | $\sigma = 20$ | 5 | 21.9 | 32.4 | 2.16 | 6.2 | 10.6 | 3.88 | 18.1 | 28.5 | 2.64 |
| | | 10 | 28.5 | 43.5 | 1.79 | 13.7 | 22.2 | 3.37 | 21.1 | 34.6 | 2.40 |
| | $\sigma = 50$ | 5 | 22.2 | 31.1 | 2.16 | 3.7 | 8.1 | 3.88 | 14.5 | 24.3 | 2.67 |
| | | 10 | 27.9 | 41.6 | 1.82 | 9.9 | 19.0 | 3.40 | 17.9 | 31.4 | 2.46 |
| User VLMs | InternVL3.5-2B | 5 | 19.5 | 30.3 | 2.19 | 7.8 | 10.4 | 4.05 | 19.0 | 27.5 | 2.77 |
| | | 10 | 26.4 | 42.2 | 1.89 | 13.0 | 20.7 | 3.49 | 19.8 | 33.1 | 2.52 |
| | Qwen2.5-VL-3B | 5 | 18.7 | 27.6 | 2.28 | 4.8 | 8.4 | 3.96 | 14.8 | 23.5 | 2.87 |
| | | 10 | 26.8 | 39.6 | 1.92 | 11.7 | 18.4 | 3.35 | 18.9 | 30.1 | 2.54 |
| Vanilla Method ($\sigma = 0$, Qwen2.5-VL-7B) | | 5 | 18.7 | 26.0 | 2.25 | 5.6 | 10.1 | 4.00 | 7.9 | 13.7 | 3.46 |
| | | 10 | 22.1 | 32.0 | 1.98 | 9.2 | 14.7 | 3.60 | 10.1 | 18.0 | 3.16 |
| CIPC ($\sigma = 0$, Qwen2.5-VL-7B) | | 5 | 24.6 | 34.3 | 1.98 | 6.2 | 10.8 | 3.76 | 19.3 | 29.8 | 2.52 |
| | | 10 | **31.3** | **45.1** | **1.62** | **14.6** | **23.5** | **3.21** | **22.8** | **36.4** | **2.24** |

## A.2 CASE STUDY

In this section, we present how our interactive multi-event video retrieval is performed. As shown in Fig. 7 and Fig. 8, the rank of the ground truth video progressively improves over interaction rounds, eventually reaching rank 1 and confirming successful retrieval.

## A.3 NOISY CASE

In this section, we present an example illustrating how noise can lead to interaction failures. As shown in Fig. 9, the noisy image causes the LLM to misinterpret the information from the frames. When asked about *"What color clothes does the toddler wear?"*, the LLM produces an incorrect response, which in turn lowers the retrieval ranking.

## A.4 ADDITIONAL ABLATION STUDIES

In this section, we present further ablation experiments to explore the effects of different design choices and experimental settings on our framework's performance.

### A.4.1 EFFECT OF USER-PROVIDED TEMPORAL HINTS

In practice, there are several ways to obtain temporal hints from users. In our experiments, we use the frame–query similarity as a lightweight, unsupervised simulation for convenience. For real users, one simple option is to let them indicate a temporal index. Specifically, users need to provide rough

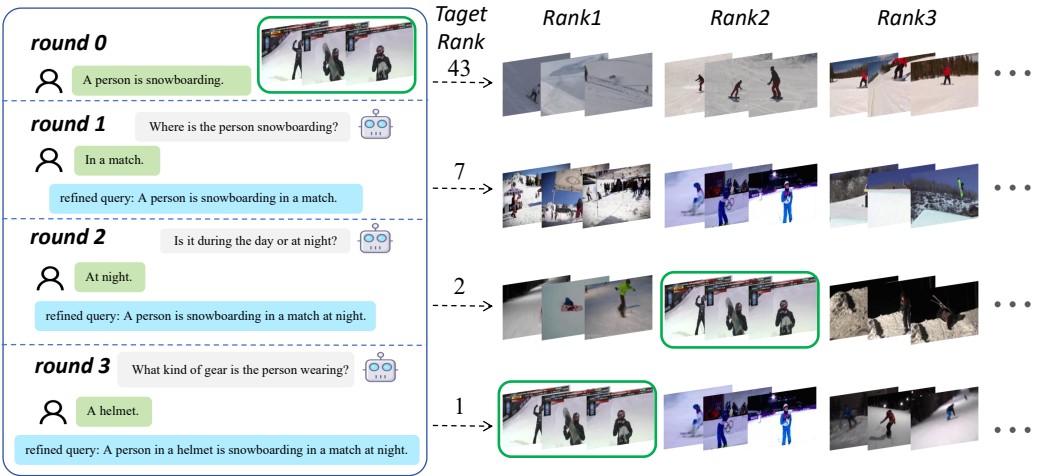

Figure 7: CIPC example 1: an interaction is conducted between the system and a user simulator on the ground truth segment. Top-3 retrievals are presented after each round.

Figure 8: CIPC example 2: an interaction is conducted between the system and a user simulator on the ground truth segment. Top-3 retrievals are presented after each round.

hints by indicating a number from 1 to 10 corresponding to the target segment in the video during interaction. Results are shown in Tab. 7. We find that even rough temporal hints can significantly improve the final retrieval results.

### A.4.2 ABLATIONS ON DIFFERENT WEIGHTING FUNCTIONS

To study the effect of position constraint, we evaluate several weighting functions, including exponential weighting, Gaussian weighting, and reciprocal weighting, which modulate the influence of temporal proximity between the query and candidate segments.

Specifically, let $p_q$ denote the query's normalized temporal position and $p_i$ denote the candidate segment's normalized temporal position. The three weighting functions are defined as follows:

- **Exponential Weighting:** $w_{\exp} = \exp(-\lambda \left| p_i - p_q \right|)$

- **Gaussian Weighting:** $w_{\text{gauss}} = \exp\left(-\frac{(p_i - p_q)^2}{2\theta^2}\right)$

- **Reciprocal Weighting:** $w_{\text{recip}} = \frac{1}{1 + \delta \left| p_i - p_q \right|}$

Figure 9: Noisy case: an example showing how noise leads to interaction errors.

Table 7: Simulation of user-provided temporal hints on ActivityNet Captions dataset. ("w" indicates the presence of temporal hints, and "w/o" represents the absence of such hints.)

| Method | R@1↑ | Hit@1↑ | BRI↓ |
|---|---|---|---|
| Temporal Index (w) | 28.6 | 43.1 | 1.65 |
| Temporal Index (w/o) | 26.0 | 41.6 | 1.70 |
| CIPC (ours) | **31.3** | **45.1** | **1.62** |

To be consistent with the weighting function used in CIPC, we set the hyperparameters in this experiment as follows: $\lambda = 0.3$ for exponential weighting, $\theta = 1.2$ for Gaussian weighting, and $\delta = 0.4$ for reciprocal weighting. Tab. 8 presents their comparison results. The performance gap among different weighting functions is relatively small.

### A.4.3 PER-ROUND LATENCY AND MEMORY ANALYSIS

To evaluate efficiency, we report per-round latency and GPU memory usage for the main components of the interactive loop, including segmentation, question generation, user response, and retrieval. We compare our method with a lightweight baseline using uniform segmentation into 8 clips. The system LLM is Qwen3-8B, and the user LLM is Qwen2.5VL-7B. Results are shown in Tab. 9 and Tab. 10. Our adaptive segmentation and context-integrated interaction introduce modest overhead while maintaining improved retrieval performance, demonstrating a reasonable trade-off between efficiency and effectiveness.

### A.4.4 ABLATION ON NUMBER OF INTERACTION ROUNDS

We conduct an ablation study to evaluate how the retrieval performance evolves as the number of interaction rounds increases. Tab. 11 shows that retrieval metrics steadily improve with more rounds, demonstrating the effectiveness of progressive refinement in our framework.

### A.4.5 ABLATION ON COARSE-TO-FINE STRATEGIES

We also conduct ablation studies to analyze different coarse-to-fine strategies. Specifically, we compare three settings: coarse-only, fine-only, and coarse-to-fine. For each query, we record the number of search rounds taken and compute the average per query. Tab. 12 shows a trade-off between efficiency and performance: coarse-only is fastest but less accurate, fine-only is most accurate but slower, while coarse-to-fine balances both, achieving near-fine accuracy with fewer rounds.

Table 8: Comparison with different weighting functions on ActivityNet Captions dataset.

| Weighting Function | R@1↑ | R@10↑ | Hit@1↑ | BRI↓ |
|---|---|---|---|---|
| Exponential Weighting | 31.2 | 69.2 | **45.2** | **1.62** |
| Gaussian Weighting | 30.2 | 68.9 | 43.3 | 1.63 |
| Reciprocal Weighting | 30.4 | **69.4** | 44.2 | 1.64 |
| CIPC (ours) | **31.3** | 69.3 | 45.1 | **1.62** |

Table 9: Per-round latency and GPU memory usage for the interactive loop on ActivityNet Captions dataset.

| Method | Segmentation | | Question Generation | | User Response | | Retrieval | |
|---|---|---|---|---|---|---|---|---|
| | Latency | Memory | Latency | Memory | Latency | Memory | Latency | Memory |
| Lightweight Baseline | **0.662s** | 1372.11MB | 0.659s | **15702.85MB** | **1.041s** | **21454.78MB** | **14.678s** | 2377.26MB |
| CIPC (ours) | 24.138s | **1336.26MB** | **0.554s** | 15703.03MB | 1.446s | 22328.46MB | 16.281s | **2265.91MB** |

### A.4.6 SENSITIVITY ANALYSIS ON CLIP BACKBONE AND FRAME SAMPLING RATE

We conduct sensitivity studies on both the CLIP backbone and the frame sampling rate $F$. For the backbone, we compare our default ViT-B/32 with ViT-B/16 and ViT-L/14. For the frame sampling rate, we evaluate $F = 16, 32, 64$ (ours), and 128. Tab. 13 and Tab.14 indicate that while the absolute performance varies with the CLIP backbone and frame sampling rate, our main focus is the improvement over the corresponding baseline. Regardless of backbone or sampling rate, applying CIPC consistently boosts retrieval metrics, demonstrating that our method effectively enhances performance across different settings.

### A.4.7 ROBUSTNESS TO A SINGLE IRRELEVANT QUESTION

We conduct an experiment to evaluate the robustness of CIPC when an unrelated question is inserted in one of the interaction rounds. Specifically, we randomly replace one round with the predefined question *"Are there any scenes involving outer space or planets?"*. Results in Tab. 15 show that this causes only a minor drop in performance, demonstrating that CIPC is robust to occasional uninformative or irrelevant queries.

### A.4.8 ABLATION OF EARLY-STOP THRESHOLDS

To evaluate the impact of early-stop criteria, we compute the similarity between consecutive refined queries. We test different thresholds to study the trade-off between efficiency and retrieval performance. As shown in Tab. 16, a lower threshold (e.g., 0.8) reduces the average rounds per query but slightly decreases retrieval accuracy. Without early stopping, metrics are highest but require more rounds. Our chosen threshold (0.9) balances accuracy and efficiency, maintaining near-optimal performance while saving interaction steps.

### A.5 PROMPT TEMPLATE

In this section, we present the prompt template for question generation, question answering, and query refinement. Fig. 10 and Fig. 11 present the prompts used by the system VLM to generate coarse-grained and fine-grained questions, respectively. Fig. 12 shows the prompt for the user VLM to answer the clarifying questions. Fig. 13 shows the prompt by the system VLM to refine queries.

## B LLM USAGE

Large Language Models (LLMs) were employed to assist in the writing and polishing of this manuscript. Specifically, an LLM was utilized to refine the language, enhance readability, and ensure clarity across various sections of the paper. Its support focused on tasks such as sentence rephrasing, grammar correction, and optimization of the overall textual flow.

Table 10: Retrieval comparison with lightweight baseline on ActivityNet Captions dataset.

| Method | R@1↑ | Hit@1↑ | BRI↓ |
|---|---|---|---|
| Lightweight Baseline | 29.6 | 43.4 | 1.68 |
| CIPC (ours) | **31.3** | **45.1** | **1.62** |

Table 11: Ablation of number of interaction rounds on ActivityNet Captions dataset.

| Rounds | R@1↑ | R@10↑ | Hit@1↑ | BRI↓ |
|---|---|---|---|---|
| 0 | 11.8 | 36.1 | 11.8 | – |
| 2 | 19.1 | 54.7 | 24.5 | 2.42 |
| 4 | 21.7 | 58.1 | 30.8 | 2.09 |
| 6 | 26.8 | 64.5 | 37.3 | 1.89 |
| 8 | 28.3 | 66.0 | 41.7 | 1.73 |
| 10 | **31.3** | **69.3** | **45.1** | **1.62** |

Notably, the LLM was not involved in ideation, research methodology design, or experimental planning. All research concepts, core ideas, and analytical work were independently developed and conducted by the authors. The LLM's contributions were limited solely to improving the linguistic quality of the paper, without any involvement in the scientific content or data analysis.

The authors assume full responsibility for the manuscript's content, including all text generated or polished with the LLM's assistance. We have verified that all text produced with the LLM's support complies with ethical guidelines and does not lead to plagiarism or any form of scientific misconduct.

Table 12: Ablation of coarse-to-fine strategies on ActivityNet Captions dataset.

| Method | R@1↑ | R@10↑ | Hit@1↑ | BRI↓ | Rounds Per Query |
|---|---|---|---|---|---|
| Coarse-only | 29.4 | 67.6 | 43.9 | 1.75 | 5.72 |
| Fine-only | **31.5** | **69.3** | **45.2** | 1.63 | 9.27 |
| Coarse-to-fine | 31.3 | **69.3** | 45.1 | **1.62** | 8.59 |

Table 13: Ablation of CLIP backbone on ActivityNet Captions dataset.

| CLIP Backbone | R@1↑ | R@10↑ | Hit@1↑ | BRI↓ |
|---|---|---|---|---|
| ViT-B/32 | 31.3 | 69.3 | 45.1 | 1.62 |
| ViT-B/16 | 33.4 | 71.1 | 45.4 | 1.54 |
| ViT-L/14 | **38.6** | **75.6** | **53.0** | **1.32** |

Table 14: Ablation of frame sampling rate $F$ on ActivityNet Captions dataset.

| Frame Sampling Rate | R@1↑ | R@10↑ | Hit@1↑ | BRI↓ |
|---|---|---|---|---|
| 16 | 23.2 | 63.7 | 34.5 | 2.14 |
| 32 | 28.8 | 66.8 | 40.9 | 1.79 |
| 64 | 31.3 | 69.3 | 45.1 | 1.62 |
| 128 | **34.9** | **76.5** | **49.3** | **1.42** |

Table 15: Robustness to a single irrelevant question in one round on ActivityNet Captions dataset.

| Method | R@1↑ | R@10↑ | Hit@1↑ | BRI↓ |
|---|---|---|---|---|
| CIPC (one round replaced by irrelevant question) | 29.8 | 68.2 | 45.0 | 1.71 |
| CIPC (original) | **31.3** | **69.3** | **45.1** | **1.62** |

Table 16: Ablation of different thresholds of early-stop criteria on ActivityNet Captions dataset. "None" indicates no early-stop.

| Threshold | R@1↑ | R@10↑ | Hit@1↑ | BRI↓ | Rounds per query |
|---|---|---|---|---|---|
| 0.8 | 29.8 | 68.1 | 44.0 | 1.72 | 7.97 |
| 0.85 | 30.5 | 68.7 | 44.3 | 1.64 | 8.17 |
| None | **31.6** | **69.5** | **45.3** | **1.60** | 10.00 |
| 0.9 (ours) | 31.3 | 69.3 | 45.1 | 1.62 | 8.59 |

---

**Coarse Question Prompt**

You are an interactive video retrieval assistant. The user provides a vague textual description of a video. Your goal is to Generate one concise clarification question that helps to identify the target segment at a COARSE LEVEL.

User's description : {QUERY}
Focus on global properties such as:
- The type of scene or environment.
- The main activity or overall event taking place.
- The presence or absence of prominent entities.

Rules for your question:
- Be specific enough to exclude other similar videos.
- Ask only ONE question at a time.
- Be concise but clear.

---

Figure 10: The prompt used for coarse-grained question generation.

---

**Fine Question Prompt**

You are an interactive video retrieval assistant. The user provides a vague textual description of a video. Your goal is to generate one concise clarification question that helps to identify the target segment at a FINE-GRAINED level.

User's description : {QUERY}
Focus on discriminative, localized details such as short actions, object interactions, or distinctive appearance attributes. Rules for your question
- The question must elicit descriptive information
- Use open formats such as 'What', 'Which', 'How', 'Where', 'Who', etc.
- Be specific enough to exclude other similar videos.
- Ask only ONE question at a time.
- Do NOT ask about timestamps, frames, or the timing of events.
- Be concise but clear.

---

Figure 11: The prompt used for fine-grained question generation.

---

**Answer Prompt**

You are a helpful assistant for answering clarifying questions. Below are the frames from a video segment related to the question based on provided video frames.

- {VIDEO_CONTENT}

- {QUESTION}

Use the visual information and question to provide a precise, accurate answer.

---

Figure 12: The prompt used to answer the question.

**Query Refinement Prompt**

You are an assistant to help improve text-video retrieval queries. Given:
The original user query:  {QUERY}
A clarifying question based on this query and retrieved video clips: {QUESTION}
The answer to that clarifying question: {ANSWER}

Please combine the original query with the information from the clarifying question and its answer to produce a refined, more specific, and informative query.

The new query should:
- Ignore trivial, redundant, or purely negative statements and retain the original intent.
- Incorporate the new details from the answer as descriptive information, not as a question.
- If the question/answer contains temporal or sequential information, rewrite the query to preserve the correct time order of events.
- Be concise but clear.
- Return ONLY the improved query as a single sentence without any additional explanation.

Figure 13: The prompt used for query refinement.

