# OpenReview forum: "Interactive Multi-event Video Retrieval with Context Integration and Position Constraint"
_ICLR.cc/2026/Conference — Submitted to ICLR 2026_

### Official Review · Reviewer_MAwe · 2025-10-27

**Soundness:** 4
**Presentation:** 4
**Contribution:** 3
**Rating:** 6
**Confidence:** 4

**Summary:**

This paper proposes **Context Integration and Position Constraint** (CIPC), a multi-round interactive video retrieval system tailored for multi-event videos.
The core contribution is threefold:

1. **Adaptive video segmentation**. This module segments videos from coarse-grained to fine-grained, thereby enabling querying at different granularities.
2. **Progressive interaction with context integration**. This module considers both segment-specific semantics as well as context-related segments, thereby producing more accurate representations for a given segment.
3. **Position constraint**. The authors additionally compute the estimated query temporal position with retrieved segments to refine temporal search accuracy.

The authors conduct many experiments on three benchmarks and show that CIPC achieves SOTA performance. The motivation of this paper is clear, with reasonable explanation of each component and promising results. I hold a positive attitude towards this paper.

**Strengths:**

1. The motivation is clearly stated and sounds reasonable.
2. Each component is rationalized with clear figures, making readers easy to follow.
3. The results are promising, and the ablation studies show that each component brings improvement.

**Weaknesses:**

1. **Unclear interaction rules.** It seems there should be a parameter to decide which rounds belong to coarse-grained search and which ones belong to fine-grained search in the Progressive Context Interaction module. But I did not find a related statement or experiments in the manuscript. Please correct me if I misunderstand something.

2. **No failure mode handling.** Since the multi-round interactive retrieval can be regarded as a decision chain, there is a possibility that there are no valid results based on the last-round query. No solutions are provided in this situation. And it is unclear what will happen if the VLM raises a totally unrelated question in one round?

3. **Reliance on strong commercial models.** From Figure 5, it seems there is a clear gap between frontier commercial models and open-source ones, indicating that the performance is dominated by the quality of generated questions. So this work may not be easy to follow for the budget-constrained research community.

**Questions:**

Since we already integrate context information into the queries, why is there still a large gap between whether using position constraint in Table 3, i.e., rows 2, 3, and 5? Does this mean the integrated context actually works for some other aspects instead of temporal alignment?

---

> ### Author Response · Authors · 2025-11-25
> **Response to Reviewer MAwe**
>
> Dear Reviewer MAwe, we thank for the helpful comments. Below we provide our responses and revisions.
>
> ---
>
> **Q1:**
> Unclear interaction rules. It seems there should be a parameter to decide which rounds belong to coarse-grained search and which ones belong to fine-grained search in the Progressive Context Interaction module. But I did not find a related statement or experiments in the manuscript. Please correct me if I misunderstand something.
>
> **A1:**
> In our Progressive Context Interaction module, we use a fixed schedule: the first 4 rounds are coarse-grained, followed by 6 fine-grained rounds, for a total of 10 rounds. This configuration balances efficiency and accuracy, allowing the model to quickly narrow down candidates in the coarse rounds and refine predictions in the fine rounds. We will clarify this design in the revision to avoid any confusion.
>
> ---
>
> **Q2:**
> No failure mode handling. Since the multi-round interactive retrieval can be regarded as a decision chain, there is a possibility that there are no valid results based on the last-round query. No solutions are provided in this situation. And it is unclear what will happen if the VLM raises a totally unrelated question in one round?something.
>
> **A2:**
> We agree that in rare cases, the final query may not match any segments, and it is also possible that the system LLM may pose a completely unrelated question during interaction. However, our framework implicitly mitigates these issues through early stop strategy and position constraint.
>
> - Early-stop strategy can automatically terminate uninformative rounds. For example, if a question is unrelated and the user’s answer does not substantially update the query, the interaction stops.
>
> - Even if one certain round introduces noise, the position constraint anchors the search within a reasonable temporal range, helping maintain retrieval accuracy.
>
> Besides, we added an experiment where we randomly insert a predefined unrelated question *"Are there any scenes involving outer space or planets?"* in one of the interaction rounds to test robustness. The results in Table 14 show that replacing one interaction round with an irrelevant question leads to only a minor drop in performance, indicating that CIPC is robust to occasional uninformative or unrelated queries.
>
> **Table 14:** Robustness to a single irrelevant question in one interaction round on ActivityNet Captions dataset.
> |Method|R@1↑|R@10↑|Hit@1↑|BRI↓|
> |-----|:---:|:----:|:------:|:---:|
> |CIPC (one round replaced by irrelevant question)|29.8|68.2|45.0|1.71|
> |CIPC(original)|**31.3**|**69.3**|**45.1**|**1.62**|
>
> ---
>
> **Q3:**
> Reliance on strong commercial models. From Figure 5, it seems there is a clear gap between frontier commercial models and open-source ones, indicating that the performance is dominated by the quality of generated questions. So this work may not be easy to follow for the budget-constrained research community.
>
> **A3:**
> Using commercial models as the system LLM generally yields better performance than open-source models, which may make it challenging for budget-constrained research community to follow. However, our main goal is to propose a general interactive retrieval framework for multi-event videos. We focus on the relative gains brought by our interaction method compared to the baseline, rather than absolute performance. In addition, we added an experiment using a more recent open-source model, Qwen3-8B, as the system LLM, and results in Table 15 indicate that the performance gap with commercial models is relatively small, showing that our method remains effective. We have made the corresponding changes in the revision.
>
> **Table 15:** Comparison of different system LLMs on Charades dataset.
> |System LLMs|R@1↑|R@10↑|Hit@1↑|BRI↓|
> |:-----:|:---:|:----:|:------:|:---:|
> |Qwen3-8B|13.0|41.6|21.8|3.49|
> |Doubao(ours)|**14.6**|**43.3**|**23.5**|**3.21**|
>
> ---
>
> **Q4:**
> Since we already integrate context information into the queries, why is there still a large gap between whether using position constraint in Table 3, i.e., rows 2, 3, and 5? Does this mean the integrated context actually works for some other aspects instead of temporal alignment?
>
> **A4:**
> The context integration mainly improves semantic understanding by incorporating information from neighboring segments, but it does not explicitly encode temporal positions. The position constraint, on the other hand, directly leverages temporal cues to align candidate segments with the query. In other words, the integrated context primarily helps with semantic refinement and disambiguation, rather than temporal alignment, which explains why adding position constraint still brings noticeable gains. Therefore, the two components address complementary aspects, semantic context and temporal alignment, and combining them yields the best performance.

---

> > ### Comment · Reviewer_MAwe · 2025-11-25
> > **Response to the rebuttal**
> >
> > Thanks for the rebuttal. All of my concerns are raised properly, and I will maintain my initial positive score.

---

### Official Review · Reviewer_nSsN · 2025-10-30

**Soundness:** 2
**Presentation:** 2
**Contribution:** 2
**Rating:** 4
**Confidence:** 4

**Summary:**

This paper proposes CIPC, a framework for interactive video retrieval over multi-event/untrimmed videos. CIPC segments videos into semantically coherent events adaptively, enables progressive multi-round interaction with context integration, and introduces a position constrain.

**Strengths:**

1. This manuscript identifies and articulates the problem of interactive retrieval in untrimmed, multi-event videos, where previous methods fall short.
2. User simulation addresses practical issues like noisy/incomplete feedback.

**Weaknesses:**

1. The related-work section surveys interactive retrieval across images/videos, but it omits several recent chat-based person re-ID papers that are highly relevant to the paper’s interactive design. Please discuss [1-3], and position your approach relative to these dialogue-driven systems (e.g., questioner design, round-level supervision).
2.  The method relies on a position constraint that re-weights segment similarity by the temporal distance between the query’s position and a segment’s center. In practice, how is the query’s temporal position obtained from a real user? The manuscript currently simulates the query position via frame–text similarity, but it is unclear how this aligns with real interactive usage where users may provide only coarse temporal hints (e.g., “near the end”).
3. The adaptive segmentation itself is clear, but please articulate the novelty beyond prior action segmentation or keyframe detection [6-8].
4. Sec. 3.3 describe a two-branch: segment-specific refinement and context-integrated refinement. But the implement details of the branches remain under-specified. Please provide a concise algorithm box or pseudo-code.
5. Please add qualitative cases where noise leads to wrong answers and analyze whether the pipeline can ever benefit from mildly incorrect answers.
6. Using the top-5 retrieved videos to condition question generation yields slight improvements but is deemed computationally inefficient. Please specify the implementation details of this strategy, quantify the extra compute, and justify the final choice not to adopt the top-5 strategy.
7. Please provide both theoretical and empirical comparisons to MERLIN [4] and UMIVR [5], with a particular focus on efficiency and scalability for long-video scenarios.

[1] Lu Y, Yang M, Peng D, et al. LLaVA-ReID: Selective Multi-image Questioner for Interactive Person Re-Identification[J]. arXiv preprint arXiv:2504.10174, 2025.
[2] Bai Y, Ji Y, Cao M, et al. Chat-based Person Retrieval via Dialogue-Refined Cross-Modal Alignment[C]//Proceedings of the Computer Vision and Pattern Recognition Conference. 2025: 3952-3962.
[3] Niu K, Yu H, Zhao M, et al. Chatreid: Open-ended interactive person retrieval via hierarchical progressive tuning for vision language models[J]. arXiv preprint arXiv:2502.19958, 2025.
[4] Han D, Park E, Lee G, et al. Merlin: Multimodal embedding refinement via llm-based iterative navigation for text-video retrieval-rerank pipeline[J]. arXiv preprint arXiv:2407.12508, 2024.
[5] Zhang B, Cao Z, Du H, et al. Quantifying and narrowing the unknown: Interactive text-to-video retrieval via uncertainty minimization[C]//Proceedings of the IEEE/CVF International Conference on Computer Vision. 2025: 22120-22130.
[6] https://katna.readthedocs.io/
[7] Kumar S, Haresh S, Ahmed A, et al. Unsupervised action segmentation by joint representation learning and online clustering[C]//Proceedings of the IEEE/CVF Conference on Computer Vision and Pattern Recognition. 2022: 20174-20185.
[8] Du Z, Wang X, Zhou G, et al. Fast and unsupervised action boundary detection for action segmentation[C]//Proceedings of the IEEE/CVF Conference on Computer Vision and Pattern Recognition. 2022: 3323-3332.

**Questions:**

1. In practice, how is the query’s temporal position obtained from real users? Since the manuscript simulates this via frame–text similarity, how will the deployed system handle coarse cues (e.g., “near the end”) or incorrect hints, and how robust is the system’s performance to such errors?
2. What is the specific novelty beyond prior action segmentation or keyframe detection (e.g., [6–8])?
3. Could you add a concise algorithmic description or pseudocode that (i) specifies when and how each branch is triggered and (ii) details the per-round operations in each branch?
4. Can you include qualitative examples in which noise leads to wrong answers, and analyze whether mildly incorrect answers can ever help the pipeline?
5. How exactly is the top-5 retrieved-video context injected into question generation? What is the additional computational cost?
6. Please provide both theoretical and empirical comparisons of efficiency and scalability (add “relative to MERLIN [4] and UMIVR [5]” for clarity).

---

> ### Author Response · Authors · 2025-11-25
> **Response to Reviewer nSsN (1/3)**
>
> Dear Reviewer nSsN, we value your thoughtful assessment of our submission and address your concerns below.
>
> ---
>
> **Q1:**
> The related-work section surveys interactive retrieval across images/videos, but it omits several recent chat-based person re-ID papers that are highly relevant to the paper’s interactive design. Please discuss and position your approach relative to these dialogue-driven systems (e.g., questioner design, round-level supervision).
>
> **A1:**
> We acknowledge that several recent chat-based person ReID works [1,2] are relevant and have now been cited in the revision. Indeed, these works have the advantage of designing very specific questions and leveraging round-level supervision, which can help efficiently discriminate highly similar candidates. However, our approach focuses on multi-event video retrieval, where designing task-specific questions is less practical due to the complex temporal and multi-event nature of videos. Instead, we adopt a general progressive interaction framework that gradually narrows down the search space from coarse to fine granularity, making it broadly applicable across different video domains.
>
> ---
>
> **Q2:**
> The method relies on a position constraint that re-weights segment similarity by the temporal distance between the query’s position and a segment’s center. In practice, how is the query’s temporal position obtained from a real user? The manuscript currently simulates the query position via frame–text similarity, but it is unclear how this aligns with real interactive usage where users may provide only coarse temporal hints (e.g., “near the end”).
>
> **A2:**
> In practice, there are several ways to obtain temporal hints from users. In our experiments, we use the frame–query similarity as a lightweight, unsupervised simulation for convenience. For real interactive usage, one simple option is to let them indicate a temporal index. Specifically, users need to provide rough hints by indicating a number from 1 to 10 corresponding to the target segment in the video during interaction. And we have also added such experiments in the revision. Results are shown in Table 1. We find that even rough temporal hints can significantly improve the final retrieval results.
>
> **Table 1:** Simulation of user-provided temporal hints on ActivityNet Captions dataset. (“w” indicates the presence of temporal hints, and “w/o” represents the absence of such hints.)
> |Method|R@1↑|Hit@1↑|BRI↓|
> |-----|:---:|:------:|:---:|
> |Temporal Index (w)  |  28.6 | 43.1  |  1.65  |
> |Temporal Index (w/o)|  26.0  |  41.6  |  1.70  |
> |CIPC(ours)          |**31.3**|**45.1**|**1.62**|
>
> Besides, we evaluate robustness to noisy hints by adding Gaussian noise to video frame features before computing query–frame similarity, so the resulting temporal hints may be inaccurate. The noise levels (10%, 20%, 50%) indicate increasing noise strength, simulating noisier or misattributed user hints. As shown in Table 2, performance decreases gradually as noise increases. However, the degradation is relatively small, demonstrating strong robustness to noisy or misattributed hints.
>
> **Table 2:** Evaluation of noisy or misattributed hints on ActivityNet Captions dataset.
> |Noise|R@1↑|R@10↑|Hit@1↑|BRI↓|
> |-----|:---:|:----:|:------:|:---:|
> |10%|31.1|**69.3**|44.7|1.64|
> |20%|30.6|68.6|44.3|1.64|
> |50%|29.3|67.6|42.2|1.70|
> |CIPC(ours)|**31.3**|**69.3**|**45.1**|**1.62**|
>
> ---
>
> **Q3:**
> The adaptive segmentation itself is clear, but please articulate the novelty beyond prior action segmentation or keyframe detection.
>
> **A3:**
> We thank the reviewer for the comment. We also surveyed prior action segmentation or keyframe detection methods, which achieve accurate boundary detection using either training or great algorithms. [3] proposes a novel unsupervised action segmentation method that jointly conducts representation learning and online clustering, introduces a temporal optimal transport module, and achieves comparable performance. However, since our framework is training-free, we use a simple, plug-and-play adaptive segmentation scheme. [4] proposes an unsupervised action boundary detection method with no training stage and low latency, yet it still relies on multi-stage processing (e.g., non-maximum suppression in local windows and subsequent clustering refinement) that requires preset parameters like action class count K (for offline scenarios) or window length, increasing deployment complexity for lightweight scenarios. In contrast, our method avoids not only training but also the need for external priors. Thus, our design is tailored to balance simplicity, efficiency, and adaptability, addressing the practical needs of lightweight action segmentation.

---

> > ### Comment · Reviewer_nSsN · 2025-11-27
> > **I maintain the overall score**
> >
> > Thank authors for the careful and detailed response. However, I have decided to maintain my overall score for the following reasons:
> > 1.I believe that interactive retrieval methods using VLMs and LLMs are closely related to this work, especially in aspects such as how to design question generation strategies. This connection is not sufficiently discussed, and a more detailed comparison with these approaches is needed.
> > 2.It is not clear that enforcing events to occur within specific temporal locations in the video is a realistic requirement in practice; users may not actually need the event to be localized to a particular segment. While leveraging such temporal constraints to improve performance is understandable, the resulting gains may primarily stem from introducing additional a priori information, rather than from a practically meaningful setting.
> > 3.The authors acknowledge that action segmentation and keyframe detection are related lines of work. In my view, these existing methods could replace the proposed adaptive video segmentation module, and they would likely achieve better performance. This diminishes the novelty of the manuscript.
> > 4.Noisy outputs are a clear and important issue that should not be downplayed. Once the large model produces incorrect descriptions or predictions, the correct candidates will obviously be adversely affected. The claimed robustness still implicitly assumes that the large model outputs are mostly correct, which may not hold for complex videos.
> > 5.The proposed top-5 retrieval conditioning strategy is still not described in enough detail, making it difficult to fully assess this part of the method.

---

> > > ### Author Response · Authors · 2025-12-01
> > > **Response to Reviewer nSsN (1/2)**
> > >
> > > We thank the reviewer nSsN for the additional comments and for clarifying the remaining concerns.
> > > Below, we address each point with further explanation.
> > >
> > > ---
> > >
> > > **Q1:**
> > > I believe that interactive retrieval methods using VLMs and LLMs are closely related to this work, especially in aspects such as how to design question generation strategies. This connection is not sufficiently discussed, and a more detailed comparison with these approaches is needed.
> > >
> > > **A1:**
> > > Our work shares similarities with VLM/LLM-based interactive retrieval methods, as both leverage LLMs to generate clarifying questions and refine user queries to improve retrieval quality. However, these methods differ fundamentally in their design motivations. Existing image-retrieval or ReID-based interactive approaches rely on task-specific and highly regularized data (e.g., pedestrians with appearance cues), allowing them to design fixed questions such as clothing color or carried objects. Likewise, most existing interactive video-retrieval works operate on pre-trimmed short clips that contain only a single event, making question generation far simpler and not requiring temporal disambiguation.
> > >
> > > In contrast, multi-event untrimmed videos contain multiple semantically diverse events and highly complex visual–temporal structures, making it difficult to design fixed or pre-defined clarifying questions. Therefore, we propose an interactive framework tailored for multi-event video retrieval. Videos are first adaptively segmented into meaningful events, after which our system performs segment-specific and context-integrated refinements and position constraint that specifically address the ambiguity and temporal complexity unique to multi-event videos.
> > >
> > > ---
> > >
> > > **Q2:**
> > > It is not clear that enforcing events to occur within specific temporal locations in the video is a realistic requirement in practice; users may not actually need the event to be localized to a particular segment. While leveraging such temporal constraints to improve performance is understandable, the resulting gains may primarily stem from introducing additional a priori information, rather than from a practically meaningful setting.
> > >
> > > **A2:**
> > > We would like to clarify that our method does not rely on precise or strong temporal supervision, nor does it assume that users must accurately localize the event.
> > >
> > > - The retrieval system still functions without any temporal hint (as shown in Table 4), demonstrating that our method does not depend on strong a priori information.
> > >
> > > - In realistic interactive scenarios, users naturally have some vague sense of when the target event occurs. Our experiments with noisy hints (Table 2) show that even highly inaccurate hints produce only minor performance drops, indicating that the model is robust to imprecise guidance.
> > >
> > > ---
> > >
> > > **Q3:**
> > > The authors acknowledge that action segmentation and keyframe detection are related lines of work. In my view, these existing methods could replace the proposed adaptive video segmentation module, and they would likely achieve better performance. This diminishes the novelty of the manuscript.
> > >
> > > **A3:**
> > > Existing action segmentation or keyframe detection methods cannot replace our adaptive segmentation module, mainly for the following reasons:
> > >
> > > - Many state-of-the-art segmentation approaches require training on video data, whether supervised, self-supervised, or unsupervised [1-3]. Our retrieval pipeline is designed to be completely training-free, so methods that rely on any form of training cannot be directly integrated.
> > >
> > > - While some methods are training-free [4], they usually rely on dataset-specific priors (e.g., number of action classes K) in offline mode, or perform online segmentation using only observed frames, lacking a global view of the entire video.
> > >
> > > Because of these fundamental mismatches in training or dataset dependence and global adaptability, existing segmentation techniques cannot directly replace our module.

---

> > > ### Author Response · Authors · 2025-12-01
> > > **Response to Reviewer nSsN (2/2)**
> > >
> > > **Q4:**
> > > Noisy outputs are a clear and important issue that should not be downplayed. Once the large model produces incorrect descriptions or predictions, the correct candidates will obviously be adversely affected. The claimed robustness still implicitly assumes that the large model outputs are mostly correct, which may not hold for complex videos.
> > >
> > > **A4:**
> > > In real interactive retrieval scenarios, it is impossible to guarantee that users always provide accurate or precise clues. Therefore, we design noise experiments to test our system’s robustness under imperfect user inputs. Specifically, we simulate two realistic types of user errors:
> > > - Noisy visual clues: We add noise to the original video segments when the user LLM answers questions based on the visual content. This models cases where users describe the video inaccurately.
> > >
> > > - Noisy temporal clues: We inject noise into frame features when computing frame–query similarity. This simulates users giving coarse or incorrect position hints.
> > >
> > > These noise settings are designed to reflect practical, real-world uncertainties during interaction. Importantly, the large model in our pipeline is not assumed to be correct. It simply serves as a proxy for human users, who may also provide incomplete or inaccurate information. Our robustness claim is therefore not based on assuming model correctness, but rather on demonstrating that CIPC maintains strong performance even when the provided clues are wrong.
> > >
> > > ---
> > >
> > > **Q5:**
> > > The proposed top-5 retrieval conditioning strategy is still not described in enough detail, making it difficult to fully assess this part of the method.
> > >
> > > **A5:**
> > > The top-5 strategy directly leverages the five most relevant video segments to generate questions that specifically discriminate among them. This makes the questions more targeted and effective in narrowing down the search space.
> > >
> > > At each interaction round, the system LLM receives the current query and the raw frames of the top-5 most relevant video segments. The prompt instructs the model: "Generate one question that clarifies ambiguous details and helps distinguish among them given a user query and 5 relevant video segments". The generated question is then answered by the user, and the answer is refined into the query for the next round.
> > >
> > > While this strategy can slightly improve retrieval performance, increasing Hit@1 from 23.5 to 24.2, it introduces significant additional computation. For example, on ActivityNet Captions, GPU latency per round rises from 0.554 seconds with our standard CIPC to 8.138 seconds, and GPU memory usage nearly doubles, increasing from 15.7GB to 30.7GB. Therefore, we adopt the simpler, more efficient design described in the manuscript.
> > >
> > > ---
> > >
> > > **References:**
> > >
> > > [1] Kumar S, Haresh S, Ahmed A, et al. Unsupervised action segmentation by joint representation learning and online clustering[C]//Proceedings of the IEEE/CVF Conference on Computer Vision and Pattern Recognition. 2022: 20174-20185.
> > >
> > > [2] S. Swetha, H. Kuehne, Y. S. Rawat and M. Shah, "Unsupervised Discriminative Embedding For Sub-Action Learning in Complex Activities," 2021 IEEE International Conference on Image Processing (ICIP), Anchorage, AK, USA, 2021, pp. 2588-2592, doi: 10.1109/ICIP42928.2021.9506759.
> > >
> > > [3] Li Z, Abu Farha Y, Gall J. Temporal action segmentation from timestamp supervision[C]//Proceedings of the IEEE/CVF Conference on Computer Vision and Pattern Recognition. 2021: 8365-8374.
> > >
> > > [4] Du Z, Wang X, Zhou G, et al. Fast and unsupervised action boundary detection for action segmentation[C]//Proceedings of the IEEE/CVF Conference on Computer Vision and Pattern Recognition. 2022: 3323-3332.

---

> ### Author Response · Authors · 2025-11-25
> **Response to Reviewer nSsN (2/3)**
>
> **Q4:**
> Sec. 3.3 describe a two-branch: segment-specific refinement and context-integrated refinement. But the implement details of the branches remain under-specified. Please provide a concise algorithm box or pseudo-code.
>
> **A4:**
> In our implementation, during the 10-round interaction process, one branch always refines the query based on the target segment alone. The other branch initially also focuses on the target segment, but after round 8, it switches to generate and answer questions regarding the preceding and succeeding segments, enabling context-integrated refinement. Below is the pseudo-code of the two-branch.
>
> ```python
> # Pseudo-code for progressive context interaction
> for round in range(10):
>     # Segment-specific refinement
>     segment_refinement.update(target_segment)
>
>     # Context-integrated refinement
>     if round <= 7:
>         context_refinement.update(target_segment)
>     else:
>         context_refinement.update(preceding_segment, succeeding_segment)
> ```
>
> ---
>
> **Q5:**
> Please add qualitative cases where noise leads to wrong answers and analyze whether the pipeline can ever benefit from mildly incorrect answers.
>
> **A5:**
> We have added qualitative case studies in the supplementary material illustrating how injected noise in user responses may lead to incorrect final retrieval. To date, we have not observed cases where the pipeline benefits from mildly incorrect answers. In most situations, noise either has no noticeable effect or slightly harms retrieval accuracy. We therefore view robustness to noise as important, but not a source of potential performance gain.
>
> ---
>
> **Q6:**
> Using the top-5 retrieved videos to condition question generation yields slight improvements but is deemed computationally inefficient. Please specify the implementation details of this strategy, quantify the extra compute, and justify the final choice not to adopt the top-5 strategy.
>
> **A6:**
> In the top-5 retrieval conditioning strategy, at each interaction round, the system LLM generates a question based on the current query and the top-5 most relevant retrieved video segments. While this approach slightly improves retrieval performance, it introduces additional computation due to the top-5 segments. We compare the latency and memory of GPU when generating questions each round with our CIPC. Table 12 shows that conditioning on the top-5 retrieved segments greatly increases cost, latency rises from 0.554s to 8.138s, and memory almost doubles (15.7GB → 30.7GB), while bringing only minor performance gains. Therefore, we choose the more efficient CIPC setting.
>
> **Table 12:** Comparison of latency and GPU memory usage of question generation per round with top-5 retrieval conditioning strategy on ActivityNet Captions dataset.
>
> |Method|Latency|Memory of GPU|
> |--------|:-------------:|:------------:|
> |Top-5 Retrieval Strategy|8.138s|30656.79MB|
> |CIPC(ours)|**0.554s** |**15703.03MB**|

---

> ### Author Response · Authors · 2025-11-25
> **Response to Reviewer nSsN (3/3)**
>
> **Q7:**
> Please provide both theoretical and empirical comparisons to MERLIN and UMIVR, with a particular focus on efficiency and scalability for long-video scenarios.
>
>
> **A7:**
> MERLIN[5] is an interactive text-video retrieval reranking framework based on LLMs. It refines query embeddings via iterative Q&A, significantly improves the performance after 5 iterations, while having limitations like insufficient temporal information modeling. UMIVR[6] is an uncertainty-minimizing interactive text-video retrieval framework that quantifies text ambiguity, mapping uncertainty, and frame uncertainty, generates targeted clarifying questions with VideoLLaVA to iteratively refine queries. However, when extended to long-video scenarios, both MERLIN and UMIVR lack effective temporal decomposition, they do not segment videos before interaction. As a result, every interaction round requires feeding the entire video into the user VLM, leading to substantial computational and memory overhead. We conducted comparative experiments and confirmed this inefficiency in practice, which is shown in Table 13. Moreover, because both frameworks rely on predefined or template-based question designs, their interaction patterns may struggle to generalize across datasets with diverse video structures and event distributions.
>
> **Table 13:** Comparison of latency and GPU memory usage of user response per round with MERLIN and UMIVR on ActivityNet Captions dataset.
>
> |Method|Latency|Memory of GPU|Rounds per round|
> |--------|:-------------:|:------------:|:------------:|
> |MERLIN|6.751s|28735.63MB|3.98|
> |UMIVR|6.847s|28735.31MB|5.00|
> |CIPC(ours)|**1.446s**|**22328.46MB**|4.13|
>
> ---
>
> **References:**
>
> [1] Lu Y, Yang M, Peng D, et al. LLaVA-ReID: Selective Multi-image Questioner for Interactive Person Re-Identification[J]. arXiv preprint arXiv:2504.10174, 2025.
>
> [2] Qin Y, Chen C, Fu Z, et al. Human-centered Interactive Learning via MLLMs for Text-to-Image Person Re-identification[C]//Proceedings of the Computer Vision and Pattern Recognition Conference. 2025: 14390-14399.
>
> [3] Kumar S, Haresh S, Ahmed A, et al. Unsupervised action segmentation by joint representation learning and online clustering[C]//Proceedings of the IEEE/CVF Conference on Computer Vision and Pattern Recognition. 2022: 20174-20185.
>
> [4] Du Z, Wang X, Zhou G, et al. Fast and unsupervised action boundary detection for action segmentation[C]//Proceedings of the IEEE/CVF Conference on Computer Vision and Pattern Recognition. 2022: 3323-3332.
>
> [5] Han D, Park E, Lee G, et al. Merlin: Multimodal embedding refinement via llm-based iterative navigation for text-video retrieval-rerank pipeline[J]. arXiv preprint arXiv:2407.12508, 2024.
>
> [6] Zhang B, Cao Z, Du H, et al. Quantifying and narrowing the unknown: Interactive text-to-video retrieval via uncertainty minimization[C]//Proceedings of the IEEE/CVF International Conference on Computer Vision. 2025: 22120-22130.

---

### Official Review · Reviewer_gFDL · 2025-10-31

**Soundness:** 3
**Presentation:** 3
**Contribution:** 3
**Rating:** 4
**Confidence:** 4

**Summary:**

The paper tackles interactive video retrieval for untrimmed, multi-event videos and proposes CIPC—a framework that (i) adaptively segments each video into event-consistent units using frame-wise similarity with an adaptive threshold (coarse→fine) (Eqs. 1–3), (ii) performs progressive interaction with context integration to refine the query with both segment-specific and neighboring-context cues, and (iii) adds a position-constraint that re-weights segment similarity by temporal proximity to the (estimated) query position, combining segment and context scores for final ranking (Eqs. 7–10). On ActivityNet Captions, Charades-STA, TVR, CIPC improves R@1 vs. prior IVR baselines and vanilla systems (e.g., up to 31.3/14.6/22.8 R@1 after 10 rounds) and beats non-interactive retrieval; ablations show each module helps and the adaptive segmentation is strongest.

**Strengths:**

Clear decomposition of the multi-event challenge into irrelevant content, context use, and position cues, with matching modules.
Method is simple, training-free over CLIP features at retrieval time, yet effective across three benchmarks with consistent R@1/Hit@1 gains and lower BRI.
Good ablations: module contributions, segmentation strategies, questioner models/strategies, and robustness via user simulation (noise and weaker VLMs).

**Weaknesses:**

1. The “query position” is simulated via frame–query similarity; clarify how real users would supply temporal hints and evaluate robustness to noisy/misattributed hints and different weighting functions.
2. Adaptive thresholding on CLIP frame features is simple; add failure-case and cross-domain analyses (e.g., rapid cuts, subtle transitions), boundary-accuracy metrics, and comparisons to learned/change-point baselines.
3. No end-to-end profiling of the interactive loop; report per-round latency/memory (segmentation, retrieval, VLM calls) on commodity CPU/GPU and compare with lightweight alternatives.
4. Lacks analysis of how question wording/granularity evolves across rounds; ablate number of turns, coarse-to-fine strategies, and stopping criteria, and quantify their impact on retrieval metrics and user effort.
5. Results depend on chosen VLMs and temperatures; report variance over prompts/seeds/backbones and include stronger multi-segment reasoning baselines (e.g., hierarchical or agentic retrieval).

**Questions:**

1. Can human users specify temporal hints (e.g., “near the end”) and how would noisy hints propagate through Eq. (8)?
2. How sensitive is CIPC to the CLIP backbone (e.g., ViT-L/14) and the frame sampling rate F?

---

> ### Author Response · Authors · 2025-11-25
> **Response to Reviewer gFDL (1/3)**
>
> Dear Reviewer gFDL, we sincerely appreciate the time you spent reviewing our work. Below we address each comment in detail.
>
> ---
>
> **Q1:**
> The “query position” is simulated via frame–query similarity; clarify how real users would supply temporal hints and evaluate robustness to noisy/misattributed hints and different weighting functions.
>
> **A1:**
> In practice, there are several ways to obtain temporal hints from users. In our experiments, we use the frame–query similarity as a lightweight, unsupervised simulation for convenience. For real users, one simple option is to let them indicate a temporal index. Specifically, users need to provide rough hints by indicating a number from 1 to 10 corresponding to the target segment in the video during interaction. And we also add such experiments in the revision. Results are shown in Table 1. We find that even rough temporal hints can significantly improve the final retrieval results.
>
> **Table 1:** Simulation of user-provided temporal hints on ActivityNet Captions dataset. (“w” indicates the presence of temporal hints, and “w/o” represents the absence of such hints.)
> |Method|R@1↑|Hit@1↑|BRI↓|
> |-----|:---:|:------:|:---:|
> |Temporal Index (w)  |  28.6 | 43.1  |  1.65  |
> |Temporal Index (w/o)|  26.0  |  41.6  |  1.70  |
> |CIPC(ours)          |**31.3**|**45.1**|**1.62**|
>
> To evaluate robustness to noisy/misattributed hints, we add Gaussian noise to video frame features before computing query–frame similarity, so the resulting temporal hints may be inaccurate. The noise levels (10%, 20%, 50%) indicate increasing noise strength, simulating noisier or misattributed user hints. As shown in Table 2, performance decreases gradually as noise increases. However, the degradation is relatively small, demonstrating strong robustness to noisy or misattributed hints.
>
> **Table 2:** Evaluation of noisy or misattributed hints on ActivityNet Captions dataset.
> |Noise|R@1↑|R@10↑|Hit@1↑|BRI↓|
> |-----|:---:|:----:|:------:|:---:|
> |10%|31.1|**69.3**|44.7|1.64|
> |20%|30.6|68.6|44.3|1.64|
> |50%|29.3|67.6|42.2|1.70|
> |CIPC(ours)|**31.3**|**69.3**|**45.1**|**1.62**|
>
> Besides, different weighting functions, such as exponential weighting, gaussian weighting, and reciprocal weighting can also be applied to modulate the influence of position constraints.
>
> Specifically, let $p_q$ denotes the query’s normalized temporal position and $p_i$ denotes the candidate segment's normalized temporal position, the three other weighting functions can be defined as follows:
>
> - **Exponential Weighting:**
> $w_\text{exp} = \exp(-\lambda \, |p_i - p_q|)$
> - **Gaussian Weighting:**
> $w_\text{gauss} = \exp\Big(-\frac{(p_i - p_q)^2}{2\theta^2}\Big)$
> - **Reciprocal Weighting:**
> $w_\text{recip} = \frac{1}{1 + \delta \, |p_i - p_q|}$
>
>
> To be consistent with the weighting function used in CIPC, we set the hyperparameters in this experiment as follows: $\lambda = 0.3$ for exponential weighting, $\theta = 1.2$ for Gaussian weighting, and $\delta = 0.4$ for reciprocal weighting. Table 3 presents their comparison results. The performance gap among different weighting functions is relatively small.
>
> **Table 3:** Comparison with different weighting functions on ActivityNet Captions dataset.
> |Weighting Fuctions|R@1↑|R@10↑|Hit@1↑|BRI↓|
> |-----|:---:|:----:|:------:|:---:|
> |Exponential Weighting|31.2|69.2|**45.2**|**1.62**|
> |Gaussian Weighting|30.2|68.9|43.3|1.63|
> |Reciprocal Weighting|30.4|**69.4**|44.2|1.64|
> |CIPC(ours)|**31.3**|69.3|45.1|**1.62**|
>
> ---
>
> **Q2:**
> Adaptive thresholding on CLIP frame features is simple; add failure-case and cross-domain analyses (e.g., rapid cuts, subtle transitions), boundary-accuracy metrics, and comparisons to learned/change-point baselines.
>
> **A2:**
> We agree that adaptive thresholding on CLIP frame features may fail in some cases, such as rapid cuts or subtle semantic transitions.
> We have also tried alternative methods, but they often require extra training or dataset-specific assumptions. For example, [1] proposes a novel unsupervised action segmentation method that jointly conducts representation learning and online clustering. [2] proposes an unsupervised action boundary detection method with no training stage, yet it still relies on multi-stage processing and requires preset parameters like action class count K (for offline scenarios) or window length.
> Since our framework is training-free, we use a simple, plug-and-play adaptive segmentation scheme.

---

> ### Author Response · Authors · 2025-11-25
> **Response to Reviewer gFDL (2/3)**
>
> **Q3:**
> No end-to-end profiling of the interactive loop; report per-round latency/memory (segmentation, retrieval, VLM calls) on commodity CPU/GPU and compare with lightweight alternatives.
>
> **A3:**
> Following this comment, we have added experiments reporting per-round latency and memory usage of GPU for the main components of the interactive loop, including segmentation, retrieval, and LLM calls. We compare our method with a lightweight baseline using uniform segmentation. In these experiments, the system LLM is Qwen3-8B, and the user LLM is Qwen2.5VL-7B. The lightweight baseline segments video frames into 8 clips uniformly. Results are shown in Table 4 and Table 5. These results demonstrate that our adaptive segmentation and context-integrated interaction introduce acceptable overhead while maintaining improved retrieval performance, showing a reasonable trade-off between efficiency and effectiveness.
>
> **Table 4:** Comparison of latency and memory usage per round with lightweight baseline on ActivityNet Captions dataset.
>
> |Method|Segmentation|Question Generation|User Response|Retrieval|
> |--------|:-------------:|:------------:|:-------------:|:------------:|
> |Lightweight Baseline|**0.662s** / 1372.11MB|0.659s / **15702.85MB**|**1.041s** / **21454.78MB**|**14.678s** / 2377.26MB|
> |CIPC(ours)|24.138s / **1336.26MB**|**0.554s** / 15703.03MB|1.446s / 22328.46MB|16.281s / **2265.91MB**|
>
>
> **Table 5:** Comparison of performance with lightweight baseline on ActivityNet Captions dataset.
>
> |Method|R@1↑|Hit@1↑|BRI↓|
> |--------|:---------:|:------:|:-------:|
> |Lightweight Baseline|29.6|43.4|1.68|
> |CIPC(ours)|**31.3**|**45.1**|**1.62**|
>
> ---
>
> **Q4:**
> Lacks analysis of how question wording/granularity evolves across rounds; ablate number of turns, coarse-to-fine strategies, and stopping criteria, and quantify their impact on retrieval metrics and user effort.
>
> **A4:**
> Unlike ReID or image retrieval, we do not design highly specific questions for each interaction, as such strategies have limited generalizability for complex video data. Instead, our goal is a general interactive approach, where the LLM progressively narrows down the search from coarse to fine granularity, allowing flexible, multi-round refinement across diverse video content.
>
> We also conduct an ablation study on the number of interaction rounds to evaluate how performance evolves with more refinement steps. The results in Table 6 show that retrieval metrics improve steadily up to a certain number of rounds.
>
> **Table 6:** Ablation of number of interaction rounds on ActivityNet Captions dataset.
> |Rounds|R@1↑|R@10↑|Hit@1↑|BRI↓|
> |:-----:|:---:|:----:|:------:|:---:|
> |0|11.8|36.1|11.8|--|
> |2|19.1|54.7|24.5|2.42|
> |4|21.7|58.1|30.8|2.09|
> |6|26.8|64.5|37.3|1.89|
> |8|28.3|66.0|41.7|1.73|
> |10|**31.3**|**69.3**|**45.1**|**1.62**|
>
> Besides, we also conduct ablation studies of coarse-to-fine strategies. To be specific, we compare three settings: coarse-only, fine-only, and coarse-to-fine. For each query, we record the number of search rounds taken under each setting. We then compute the average number of rounds per query for each strategy. Table 7 shows a trade-off between efficiency and performance. Coarse-only is fastest (5.72 rounds) but less accurate, fine-only is most accurate (R@1 31.5) but slower (9.27 rounds), while coarse-to-fine balances both, achieving near-fine accuracy (R@1 31.3) with fewer rounds (8.59).
>
> **Table 7:** Ablation of coarse-to-fine strategies on ActivityNet Captions dataset.
> |Method|R@1↑|R@10↑|Hit@1↑|BRI↓|Rounds Per Query
> |-----|:---:|:----:|:------:|:---:|:---:|
> |Coarse-only|29.4|67.6|43.9|1.75|5.72|
> |Fine-only|**31.5**|**69.3**|**45.2**|1.63|9.27|
> |Coarse-to-fine|31.3|**69.3**|45.1|**1.62**|8.59|
>
> For the early-stop criteria, we calculate the similarity of consecutive refined queries. We also conduct ablations of different thresholds. Results are shown in Table 8. A lower threshold (e.g., 0.8) reduces the average rounds per query (7.97) but slightly lowers retrieval performance, while no early stopping (“None”) achieves the best metrics (R@1 31.6) at the cost of more rounds (10). Our chosen threshold (0.9) strikes a balance, maintaining near-optimal accuracy (R@1 31.3) while saving computation (8.59 rounds per query).
>
> **Table 8:** Ablation of different thresholds of early-stop criteria on ActivityNet Captions dataset. “None” indicates that no early-stop is applied.
> |Threshold|R@1↑|R@10↑|Hit@1↑|BRI↓|Rounds per query
> |:-----:|:---:|:----:|:------:|:---:|:---:|
> |0.8|29.8|68.1|44.0|1.72|7.97|
> |0.85|30.5|68.7|44.3|1.64|8.17|
> |None|**31.6**|**69.5**|**45.3**|**1.60**|10.00|
> |0.9(ours)|31.3|69.3|45.1|1.62|8.59|

---

> ### Author Response · Authors · 2025-11-25
> **Response to Reviewer gFDL (3/3)**
>
> **Q5:**
> Results depend on chosen VLMs and temperatures; report variance over prompts/seeds/backbones and include stronger multi-segment reasoning baselines (e.g., hierarchical or agentic retrieval).
>
> **A5:**
> We follow the suggestion and evaluate our method across different prompts and random seeds. Specifically, we designe three prompt styles, simple, moderate (ours), and complex, and run each configuration with multiple random seeds (42, 123, 2025). The results in Table 9 show low variance, indicating that our approach is stable with respect to prompt wording and initialization.
>
> **Table 9:** Ablation of different styles of prompts on ActivityNet Captions dataset.
> |Prompts Styles|R@1↑|R@10↑|Hit@1↑|BRI↓|
> |:-----:|:---:|:----:|:------:|:---:|
> |Simple|31.0±0.15|68.9±0.12|44.9±0.18|1.64±0.02|
> |Complex|**31.4**±0.13|69.2±0.10|**45.3**±0.15|**1.61**±0.01|
> |Moderate(ours)|31.3±0.12|**69.3**±0.11|45.1±0.16|1.62±0.01|
>
> ---
>
> **Q6:**
> How sensitive is CIPC to the CLIP backbone (e.g., ViT-L/14) and the frame sampling rate F?
>
> **A6:**
> We conduct additional sensitivity studies on both the CLIP backbone and the frame sampling rate F. For the backbone, we compare our default ViT-B/32 with ViT-L/14 and ViT-B/16. For the sampling rate, we evaluate F = 16, 32, 64 (ours), and 128. Tables 10 and 11 indicate that while the absolute performance varies with the CLIP backbone and frame sampling rate, our main focus is the improvement over the corresponding baseline. Regardless of backbone or sampling rate, applying CIPC consistently boosts retrieval metrics, demonstrating that our method effectively enhances performance across different settings.
>
> **Table 10:** Ablation of CLIP backbone on ActivityNet Captions dataset.
>
> |CLIP Backbone|R@1↑|R@10↑|Hit@1↑|BRI↓|
> |:-----:|:---:|:----:|:------:|:---:|
> |ViT-B/32|31.3|69.3|45.1|1.62|
> |ViT-B/16|33.4|71.1|45.4|1.54|
> |ViT-L/14|**38.6**|**75.6**|**53.0**|**1.32**|
>
>
> **Table 11:** Ablation of frame sampling rate F on ActivityNet Captions dataset.
> |Frame Sampling Rate|R@1↑|R@10↑|Hit@1↑|BRI↓|
> |:-----:|:---:|:----:|:------:|:---:|
> |16|23.2|63.7|34.5|2.14|
> |32|28.8|66.8|40.9|1.79|
> |64|31.3|69.3|45.1|1.62|
> |128|**34.9**|**76.5**|**49.3**|**1.42**|
>
> ---
>
> **References:**
>
> [1] Kumar S, Haresh S, Ahmed A, et al. Unsupervised action segmentation by joint representation learning and online clustering[C]//Proceedings of the IEEE/CVF Conference on Computer Vision and Pattern Recognition. 2022: 20174-20185.
>
> [2] Du Z, Wang X, Zhou G, et al. Fast and unsupervised action boundary detection for action segmentation[C]//Proceedings of the IEEE/CVF Conference on Computer Vision and Pattern Recognition. 2022: 3323-3332.

---

### Official Review · Reviewer_UAiG · 2025-11-01

**Soundness:** 3
**Presentation:** 3
**Contribution:** 2
**Rating:** 6
**Confidence:** 5

**Summary:**

This paper proposes a  framework for Interactive Video Retrieval (IVR) in the challenging and realistic scenario of multi-event (untrimmed) videos. The authors identify the limitations of prior IVR methods, which primarily focus on pre-trimmed videos, and articulate three key challenges: sensitivity to irrelevant content, lack of context exploitation, and insufficient position exploration. CIPC addresses these issues through three main components: an Adaptive Video Segmentation strategy, Progressive Interaction with Context Integration, and a Position Constraint mechanism. The empirical results demonstrate significant improvements over both interactive and non-interactive baselines, with ablation studies validating the efficacy of each proposed component.

**Strengths:**

1.The motivation is clear.
2. The paper is well-presented

**Weaknesses:**

1.While applying Interactive Retrieval, a recent hot topic accelerated by Large Language Models (LLMs), to the domain of untrimmed video is a good idea, the paper lacks a sufficiently deep theoretical consideration on the optimal interactive query strategy for Video Retrieval. In established Interactive Retrieval fields like Person Re-Identification (ReID) or image retrieval, the primary research focus is on determining what types of questions the LLM should ask to effectively discriminate between highly similar candidates (e.g., asking about fine-grained attributes of two similar-looking individuals). For Video Retrieval, where video events are complex and span time, the core challenge of the interaction remains ambiguous. The authors highlight the limitations of existing Interactive Retrieval methods but fail to clearly articulate the unique, high-level attributes (e.g., subtle temporal differences, inter-event causal relations) that the model should target during progressive questioning. This oversight suggests the framework may be relying on a generic LLM interaction, which is sub-optimal for unlocking the full potential of interactive retrieval in complex temporal sequences.

2. The proposed Adaptive Video Segmentation relies on an adaptive threshold ($\tau$) calculated from the magnitude of visual feature distance ($\delta_t$) between adjacent frames. This strategy is fundamentally susceptible to scenarios where visual dynamics decouple from semantic coherence, leading to critical segmentation failures:Case A: Semantic Shift with Visual Smoothness: A major semantic boundary occurs (e.g., the completion of one complex sub-task and the initiation of the next) under a visually continuous camera operation (e.g., slow pan, zoom, or long take). The resulting small $\delta_t$ will likely cause the two distinct semantic events to be erroneously merged into a single segment, thus misaligning the available retrieval unit with the user's target.Case B: Visual Jitter with Semantic Coherence: A single, semantically continuous event (e.g., a conversation or continuous action) is recorded using rapid, visually jarring techniques like quick cuts or extreme hand-held motion. The resultant large, spurious $\delta_t$ will cause the event to be incorrectly fragmented into multiple, artificially short segments.The reliance on a purely visual-distance metric, even with an adaptive threshold, suggests a vulnerability in the system's foundational step, posing a significant risk of propagating initial segmentation errors to the subsequent, refinement-based retrieval stages.

3.The Context Integration mechanism, which refines the query by utilizing information exclusively from the immediate preceding and succeeding video segments, enforces a strong Markovian assumption on the inter-event dependencies. While effective for short-range relationships, this local scope inherently limits the framework's ability to resolve target events that require long-range temporal or causal context. In narrative-heavy or procedural long videos (e.g., documentaries, instructional content), the most salient contextual cues might be located far from the target event. This design choice, therefore, places a theoretical ceiling on the complexity of video relationships the model can effectively leverage, preventing it from tackling the full breadth of real-world multi-event video understanding.

4. Related paper to be cited: IVCR-200K: A LARGE-SCALE MULTI-TURN DIALOGUE BENCHMARK FOR INTERACTIVE VIDEO CORPUS RETRIEVAL

**Questions:**

na

---

> ### Author Response · Authors · 2025-11-25
> **Response to Reviewer UAiG (1/2)**
>
> Dear Reviewer UAiG, thank you for your constructive and helpful comments. Below we summarize how we have addressed your concerns.
>
> ---
>
> **Q1:**
> While applying Interactive Retrieval, a recent hot topic accelerated by Large Language Models (LLMs), to the domain of untrimmed video is a good idea, the paper lacks a sufficiently deep theoretical consideration on the optimal interactive query strategy for Video Retrieval. In established Interactive Retrieval fields like Person Re-Identification (ReID) or image retrieval, the primary research focus is on determining what types of questions the LLM should ask to effectively discriminate between highly similar candidates (e.g., asking about fine-grained attributes of two similar-looking individuals). For Video Retrieval, where video events are complex and span time, the core challenge of the interaction remains ambiguous. The authors highlight the limitations of existing Interactive Retrieval methods but fail to clearly articulate the unique, high-level attributes (e.g., subtle temporal differences, inter-event causal relations) that the model should target during progressive questioning. This oversight suggests the framework may be relying on a generic LLM interaction, which is sub-optimal for unlocking the full potential of interactive retrieval in complex temporal sequences.
>
> **A1:**
> LLaVA-ReID [1] is a selective multi-image questioner based on multimodal models, designed for interactive person re-identification tasks. By integrating selective visual context and looking-forward supervision strategies, LLaVA-ReID can generate targeted questions to iteratively refine the initial vague descriptions provided by witnesses. [2] proposes an MLLM-driven Interactive Cross-modal Learning framework, enhances text queries via multi-round interactions and enriches training data diversity, achieving remarkable improvements in performance.
> While these works demonstrate the effectiveness of carefully designed question strategies in ReID and image retrieval, such task-specific question formulation is often difficult to generalize beyond their domains.
> In contrast, our work aims to provide a more general interaction framework for untrimmed video retrieval. Instead of manually defining question types, we introduce context-integrated progressive interaction and position constraints to handle multi-event, temporally complex videos in a training-free, plug-and-play manner.
>
> ---
>
> **Q2:**
> The proposed Adaptive Video Segmentation relies on an adaptive threshold ($\tau$) calculated from the magnitude of visual feature distance ($\delta_t$) between adjacent frames. This strategy is fundamentally susceptible to scenarios where visual dynamics decouple from semantic coherence, leading to critical segmentation failures: Case A: Semantic Shift with Visual Smoothness: A major semantic boundary occurs (e.g., the completion of one complex sub-task and the initiation of the next) under a visually continuous camera operation (e.g., slow pan, zoom, or long take). The resulting small $\delta_t$ will likely cause the two distinct semantic events to be erroneously merged into a single segment, thus misaligning the available retrieval unit with the user's target. Case B: Visual Jitter with Semantic Coherence: A single, semantically continuous event (e.g., a conversation or continuous action) is recorded using rapid, visually jarring techniques like quick cuts or extreme hand-held motion. The resultant large, spurious $\delta_t$ will cause the event to be incorrectly fragmented into multiple, artificially short segments.The reliance on a purely visual-distance metric, even with an adaptive threshold, suggests a vulnerability in the system's foundational step, posing a significant risk of propagating initial segmentation errors to the subsequent, refinement-based retrieval stages.
>
> **A2:**
> We agree that both cases may occur. These cases highlight inherent challenges of segmenting untrimmed videos. We also try alternative approaches such as action segmentation methods, but they often rely on dataset-specific assumptions or require additional training.
> [3] proposes a novel unsupervised action segmentation method that jointly conducts representation learning and online clustering, introduces a temporal optimal transport module, and achieves comparable performance. [4] proposes an unsupervised action boundary detection method with no training stage and low latency, yet it still relies on multi-stage processing and requires preset parameters like action class count K (for offline scenarios) or window length, increasing deployment complexity for lightweight scenarios.
> Since our framework is training-free, we use a simple, plug-and-play adaptive segmentation scheme. In future work, we plan to improve this module, for example by adding Gaussian smoothing to reduce noise and make the segmentation more robust.

---

> ### Author Response · Authors · 2025-11-25
> **Response to Reviewer UAiG (2/2)**
>
> **Q3:**
> The Context Integration mechanism, which refines the query by utilizing information exclusively from the immediate preceding and succeeding video segments, enforces a strong Markovian assumption on the inter-event dependencies. While effective for short-range relationships, this local scope inherently limits the framework's ability to resolve target events that require long-range temporal or causal context. In narrative-heavy or procedural long videos (e.g., documentaries, instructional content), the most salient contextual cues might be located far from the target event. This design choice, therefore, places a theoretical ceiling on the complexity of video relationships the model can effectively leverage, preventing it from tackling the full breadth of real-world multi-event video understanding.
>
> **A3:**
> We agree that some videos may rely on long-range context, and using only nearby segments cannot capture all such cases. Our framework, however, focuses on interactive retrieval, not full video understanding. We assume users can supply short, nearby clues that help narrow the search. Whether those surrounding segments are perfectly related to the target is not critical.
>
> ---
>
> **Q4:**
> Related paper to be cited: IVCR-200K: A LARGE-SCALE MULTI-TURN DIALOGUE BENCHMARK FOR INTERACTIVE VIDEO CORPUS RETRIEVAL
>
> **A4:**
> Thanks for pointing out this relevant work. After reviewing the paper, we agree that it is highly related, and we have added the citation in the revision.
>
> ---
>
> **References:**
>
> [1] Lu Y, Yang M, Peng D, et al. LLaVA-ReID: Selective Multi-image Questioner for Interactive Person Re-Identification[J]. arXiv preprint arXiv:2504.10174, 2025.
>
> [2] Qin Y, Chen C, Fu Z, et al. Human-centered Interactive Learning via MLLMs for Text-to-Image Person Re-identification[C]//Proceedings of the Computer Vision and Pattern Recognition Conference. 2025: 14390-14399.
>
> [3] Kumar S, Haresh S, Ahmed A, et al. Unsupervised action segmentation by joint representation learning and online clustering[C]//Proceedings of the IEEE/CVF Conference on Computer Vision and Pattern Recognition. 2022: 20174-20185.
>
> [4] Du Z, Wang X, Zhou G, et al. Fast and unsupervised action boundary detection for action segmentation[C]//Proceedings of the IEEE/CVF Conference on Computer Vision and Pattern Recognition. 2022: 3323-3332.

---

### Comment · Area_Chair_n1pX · 2025-11-25

Dear Reviewers,

Thank you for your time and effort in reviewing submissions for ICLR 2026. As we begin the author-reviewer discussion process, we kindly remind you to submit your responses to the author rebuttals by **December 2**.

Your engagement in this discussion phase is crucial to ensuring a fair and thorough evaluation of each submission.

### **Action Required**
- Carefully consider the authors’ rebuttal and any additional evidence they provide.
- Update your review (if applicable) to reflect your revised perspective.
- Discuss with the authors if further details are required

Your AC

---

### Author Response · Authors · 2025-12-02
**Summary of Reviews to the Area Chair**

Dear AC,

Here we would like to summarize how we have addressed the primary concerns raised by reviewers. We have conducted a wide range of new experiments.

**1. Comparison with Prior Interactive Retrieval Work**

Reviewers **UAiG** and **nSsN** noted that related works on chat-based person ReID were not fully discussed. We added relevant citations and clarified that existing methods focus on single-event or task-specific settings with fixed questions. In contrast, our method targets multi-event, untrimmed videos with complex temporal structures, using adaptive segmentation, progressive coarse-to-fine interaction, context integration, and position constraint, making it broadly applicable across video domains.

**2. Adaptive Segmentation Module**

Reviewers **UAiG** and **gFDL** commented that adaptive segmentation may appear simple and may struggle with certain complex scenarios. We clarified that the goal of this module is not precise action segmentation, but lightweight, training-free event partitioning suitable for interactive retrieval. Its simplicity is intentional to ensure generality and efficiency across diverse multi-event videos. Reviewer **nSsN** questioned whether existing action segmentation or keyframe detection methods could replace our module. We explained that most state-of-the-art segmentation methods require training, while training-free alternatives often rely on dataset-specific priors. These mismatches make them incompatible with our training-free interactive retrieval pipeline.

**3. Simulation of Temporal Hints and Noise Robustness**

Reviewers **gFDL** and **nSsN** raised concerns about how temporal information is obtained from real users and whether the system remains reliable when users provide inaccurate hints. We simulate realistic user guidance by letting users provide a coarse segment index. Moreover, we conducted noise experiments on both visual and temporal hints to validate robustness.

**4. Additional Ablation Studies and Implementation Details**

Reviewers **gFDL** and **MAwe** raised questions about other design choices and hyperparameters in our framework. We performed ablations on position constraint weighting functions, interaction rounds, interaction strategies, backbone choice, frame sampling, and so on. We also measured memory usage and latency for each component to verify efficiency.

We hope this summary offers a clear perspective on our work and sincerely thank you for your time and thoughtful review.

Sincerely,
Authors of Submission2748

---

### Author Response · Authors · 2025-12-02
**Summary Comment to the Area Chair**

Dear AC,

We sincerely appreciate your time and effort in the review process. After the rebuttal phase, Reviewer MAwe confirmed that all their concerns were fully resolved and maintained their positive score. Reviewer nSsN raised additional questions after reading our rebuttal, and we provided a detailed follow-up response, clarifying the misunderstanding and addressing the issue thoroughly. Unfortunately, due to an unexpected event that caused the discussion phase to end early, we were unable to receive replies from Reviewer UAiG or Reviewer gFDL.

Thank you again for your careful consideration.

Sincerely,
Authors of Submission2748

---

### Meta-Review · Area_Chair_fsZd · 2026-01-04

**Summary:**

While the reviewers acknowledged the practical motivation of addressing interactive retrieval for multi-event videos, the decision to reject is driven by significant concerns regarding the technical novelty and the realism of the proposed framework. Specifically, the adaptive segmentation module was viewed by multiple reviewers as incremental or replaceable by existing methods. Furthermore, there are persistent doubts about whether the position constraint and the underlying interaction assumptions realistically reflect how users provide temporal hints in real-world scenarios.

**Reviewer Concerns:**

The authors successfully addressed concerns regarding computational efficiency and implementation details. However, major concerns remain outstanding. Reviewer nSsN and UAiG noted that the novelty of the training-free segmentation is limited compared to established action segmentation baselines. Additionally, the theoretical depth of the question-generation strategy was considered insufficient compared to recent dialogue-based ReID works, and the robustness of the system relies on assumptions about user inputs that may not hold in complex, open-ended interactions.

**Reviewer Scores:**

Reviewer MAwe explicitly confirmed satisfaction with the rebuttal and would maintain their positive score 6. Reviewer nSsN actively participated in the discussion and explicitly stated they would maintain their rejection score 4 due to unresolved concerns about novelty and interaction design. Reviewers UAiG and gFDL did not respond to the final rebuttal; however, given that the core issues regarding the realism of user simulation and segmentation novelty were echoed by nSsN, it is likely they would have maintained their respective scores 6 and 4 or shifted slightly downwards upon realizing the limitations in novelty.

---

### Decision · Program_Chairs · 2026-01-26

Reject